# Dual Latent Memory for Visual Multi-agent System

Xinlei Yu [1]   Chengming Xu [2]   Zhangquan Chen [3]   Bo Yin [1]   Cheng Yang [4]   Yongbo He [5]   Yihao Hu [6]
Jiangning Zhang [5]   Cheng Tan [7]   Xiaobin Hu [1]   Shuicheng Yan [1]

## Abstract

While Visual Multi-Agent Systems (VMAS) promise to enhance comprehensive abilities through inter-agent collaboration, empirical evidence reveals a counter-intuitive "scaling wall": increasing agent turns often degrades performance while exponentially inflating token costs. We attribute this failure to the information bottleneck inherent in text-centric communication, where converting perceptual and thinking trajectories into discrete natural language inevitably induces semantic loss. To this end, we propose **L²-VMAS**, a novel model-agnostic framework that enables inter-agent collaboration with dual latent memories. Furthermore, we decouple the perception and thinking while dynamically synthesizing dual latent memories. Additionally, we introduce an entropy-driven proactive triggering that replaces passive information transmission with efficient, on-demand memory access. Extensive experiments among backbones, sizes, and multi-agent structures demonstrate that our method effectively breaks the "scaling wall" with superb scalability, improving average accuracy by *2.7-5.4%* while reducing token usage by *21.3-44.8%*.

## 1. Introduction

The rapid development of vision-language models (VLMs) has significantly advanced visual perception and thinking capabilities. Consequently, research is shifting from a single-agent paradigm to a VMAS framework to leverage collective intelligence (Lei et al., 2025; Yu et al., 2025c). The core hypothesis is that VMAS, through more agent turns of collaboration, can yield performance gains over the single models. Ideally, deeper agent turns allow agents to obtain more accurate perceptual observations and superior thinking trajectories, expected to achieve greater robustness and accuracy through multi-agent perspectives. However, unlike the envisioned positive scaling observed in text-based multi-agent collaboration (Qian et al., 2025; Kim et al., 2025), transitioning to the visual system poses distinct challenges.

We attribute this failure to the limitations of the ***text-centric*** inter-agent communication paradigm, as illustrated in Figure 1. Theoretically, transmitting information solely via natural language creates a significant ***information bottleneck***, especially in multimodal models (West et al., 2019; Huang et al., 2025b). While natural language compresses and decodes internal state of the model into human-readable discrete tokens, this process is inherently inefficient and lossy (Wang & Utiyama, 2024). Thus, it cannot be perfectly used as a means of inter-agent communication to be efficient, accurate, and robust (Malfa et al., 2025; Cemri et al., 2025). Especially in VMAS, where textual communication is the medium, dense visual perceptual observations is converted into textual descriptions, resulting in the loss of fine-grained details, and the omission of hidden cognitive states and intrinsic semantic details results in inferior thought exchange among agents. Additionally, the perception and thinking will be conflated and decoded into discrete tokens by previous agents, and then encoded by downstream agents. As inter-agent interaction continues, these semantic losses accumulate, and substantial duplicate tokens serve as the context of agents, leading to undesirable performance degradation and computation explosion. Our empirical investigation identifies this "***scaling wall***" of VMAS: simply increasing agent turns fail to guarantee improvement.

From the preliminary analysis results (see in Figure 2), the existing VMAS exhibits notable ineffectiveness as the number of agent turns increases. The task accuracy of the multi-agent systems exhibits a non-monotonic variation trend with the increase of interaction turns: it reaches a marginal peak at the 3*rd* turn, then undergoes continuous attenuation, and even falls below the single-agent baseline by the 6*th* turn. Meanwhile, the computational overhead of the system rises sharply as the interaction proceeds; specifically, the total token consumption surges by more than 30 times by the 10*th* turn. This phenomenon reveals a critical trade-off dilemma between agent-wise scaling, computational cost, and task

---
[1]NUS   [2]FDU   [3]THU   [4]DeepWisdom   [5]ZJU   [6]HNU
[7]Shanghai AI Lab.   Correspondence to:   Xiaobin Hu
<ben0xiaobin0hu1@nus.edu.sg>.

*Proceedings of the 43ʳᵈ International Conference on Machine Learning*, Seoul, South Korea. PMLR 306, 2026. Copyright 2026 by the author(s).

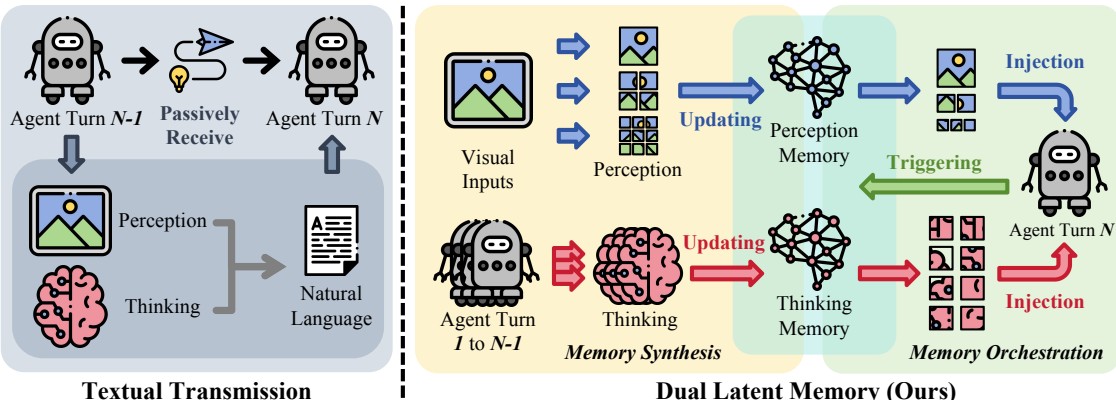

*Figure 1.* Comparison with existing text-based information transmission, and our proposed dual latent memory tailored for VMAS.

performance in multi-agent collaboration frameworks. As illustrated in Figure 3, the coupling of perception and thinking trajectories is also the culprit of performance decline, which is more severe in deeper agent turns. It induces mutual interference and disregards their inherently distinct functional attributes, even underperforming the method that only relays the conclusion.

To address these limitations, we propose **L²-VMAS**, a model-agnostic framework that shifts the collaboration medium from natural language to dual latent memories. It consists of two separate parts: memory synthesis and memory orchestration. For the former, we construct a dual latent memory shared by all agents that decouples memories into *perception* and *thinking*, which are *dynamically synthesized* by all previous agent turns. For the latter, we introduce shifting the information communication from passive reception to *proactive orchestration*, which uses entropy-based signals to retrieve the two memories only when necessary. By adopting this dual, latent, and proactive paradigm, we overcome the "scaling wall" and enable efficient multi-agent collaboration in visual scenarios.

We validate our proposed method across five VLM backbones, four model sizes, and six multi-agent structures. The results demonstrate that our framework significantly improves average accuracy while reducing token consumption, with superior dynamicity, proactivity, and scalability. Our contributions are summarized as follows:

- **Failure Analysis of VMAS.** We quantitatively characterize the "scaling wall" in VMAS, identifying the text-centric information bottleneck and failed coupling of perception and thinking.

- **Dual Latent Memory Synthesis.** We introduce a shared and dynamically synthesized memory system that decouples perception and thinking, enabling effective information interaction without interference.

- **Proactive Memory Orchestration.** We propose an

entropy-driven memory triggering and injection mechanism, transforming passive collaboration into an efficient, on-demand process.

## 2. Requisite Analyses

In VMAS, each turn of interaction takes the perceptual observations and cognitive thinking trajectories generated by preceding agents as core inputs. While textual natural language is widely adopted as a direct communication medium for inter-agent information transmission, it inevitably induces non-negligible information loss and semantic bias during text encoding and decoding. These issues are further exacerbated by the unstructured mixing of perception and thinking, which leads to severe information interference and thus impairs the reliability of subsequent agents. To conduct a rigorous quantitative evaluation, we perform experiments on MMBench (Liu et al., 2024) benchmark using Qwen3-VL-8B-Thinking (Bai et al., 2025). Supplementary details and more results are available in Appendix B.

### 2.1. Analytical Experiments

**Accuracy and Token Usage Across Agent Turns.** To pinpoint the inherent limitations of existing VMAS, we begin by quantitatively measuring two core metrics across distinct agent turns: task accuracy and total token usage. The former metric quantifies the effectiveness and performance, while the latter reflects the integrated computational and communicative efficiency. When the agent turn is 1, the system is equivalent to an independent model without inter-agent collaboration, serving as the single-agent baseline.

As illustrated in Figure 2 (complete results in Figure 7), a clear trend emerges as the number of agent turns increases: the task accuracy exhibits a pattern of marginal initial ascent followed by a precipitous decline, whereas the total token consumption surges progressively with an ever-accelerating growth rate. Specifically, the accuracy peaks at the 3*rd* agent turn, rising modestly from 84.8 to 86.6, for a mere 2.1%

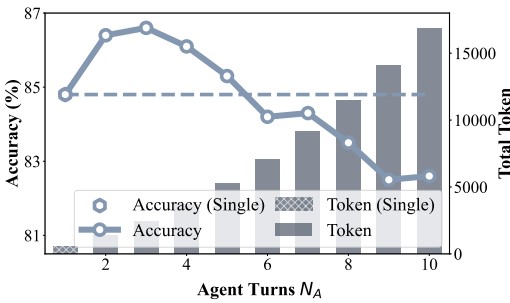

*Figure 2.* Accuracy and total token usage among agent turns.

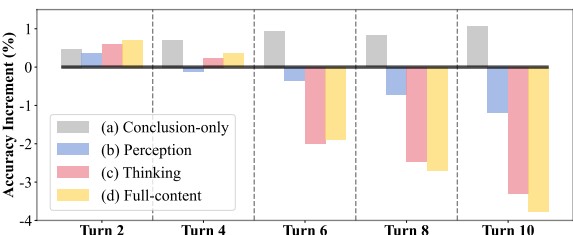

*Figure 3.* Comparison of inter-agent transmission content.

relative improvement over the initial turn. Thereafter, the accuracy embarks on a sustained downward trajectory, falling below the performance of the single-agent baseline starting with the 6*th* turn and even lower by 2.6% at the 10*th* turn. Furthermore, while the total token consumption averages a mere 557 tokens in the single-agent setting, it escalates sharply to 5,241 tokens at the 5*th* turn and a staggering 16,840 tokens at the 10*th* turn, representing more than 9-fold and 30-fold increases relative to the single-agent baseline, respectively. Such pronounced performance degradation and exponential computational overhead collectively erode the reliability and scalability of VMAS when deployed in complex multi-agent scenarios and real-world tasks.

**Inter-agent Transmission Content.** Building on prior empirical findings (Zheng et al., 2025; Fu et al., 2025; Zou et al., 2025), text-centric inter-agent communication paradigms are considered as a primary culprit behind the dual predicaments of task performance degradation and computational efficiency loss in LLM-based multi-agent systems. However, in VMAS, the integration and transmission of visual-perceptual information exacerbate these predicaments. Against this backdrop, we seek to delve deeper into the root causes by systematically comparing accuracy across varying inter-agent transmission content. To this end, we set four groups with different inter-agent transmission content: (a) Conclusion-only: solely transmits the conclusion of each agent, omitting intermediate details; (b) Perception: transmits the perceptual observations and conclusion; (c) Thinking: transmits cognitive thinking and the conclusion; (d) Full-content: transmits complete outputs. Detailed settings are shown in Appendix B.1.

As presented in Figure 3 (complete results in Figure 8), a counterintuitive pattern emerges: in early-stage agent turns, richer contextual information derived from preceding agents yields a measurable boost in performance; by contrast, in later-stage agent turns, the unregulated accumulation and conflation of unstructured perceptual observations and cognitive thinking trajectories give rise to substantial performance degradation. To elaborate, the Conclusion-Only transmission paradigm delivers steady improvements of 0.5% to 1.1% across successive turns. In stark contrast, while the Full-content transmission achieves a marginal 0.7% accuracy gain at the 2*nd* turn, it incurs a net 3.8% performance loss by the 10*th* turn, underperforming both the Perception and Thinking transmission configurations. These findings conclusively affirm that the prevailing text-centric inter-agent communication paradigm is inherently plagued by lossy information redundancy and severe information interference between perception and thinking.

## 2.2. Findings

Based on these experiments and analyses, two core findings are summarized as follows:

- Existing VMAS suffer from performance degradation and exponential token overhead in deep agent turns, compromising their reliability and scalability.

- Existing text-centric inter-agent interaction paradigm of VMAS, which conflates perception and thinking, proves inferior as a transmission medium.

## 3. Methodologies

### 3.1. Preliminary

**Motivation.** Based on the requisite analyses, we distill the core drawbacks of current VMAS as the primary motivation for our work: **(1) Efficiency:** Inter-agent natural language information transmission is inherently inefficient and information-lossy, and is prone to the unstructured conflation of perception and thinking trajectories; **(2) Proactivity:** Current inter-agent information transmission is rigidly triggered, which passively receives information from its predecessor prior to initiating; **(3) Scalability:** Inter-agent communication overhead escalates exponentially with the number of agent turns, leading to pronounced performance degradation and inferior scalability. To this end, we propose a novel latent dual memory system with targeted designs: dual latent memory decoupling (efficiency), an active entropy-triggered mechanism (proactivity), and dynamic memory synthesis (scalability) to enhance performance and reduce token consumption relative to the existing VMAS.

**Modeling and Formulation.** We first model the VMAS, which is composed of a set of VLM-based agents denoted

as: $\mathcal{S}_{\mathcal{A}} = \{\mathcal{A}_1, \ldots, \mathcal{A}_{N_A}\}$, where $N_A$ denotes the number of agent turns. Each agent $\mathcal{A}_i$ is instantiated with a VLM as its backbone. The connectivity of the VMAS is defined by a directed graph $\mathcal{G} = (\mathcal{S}_{\mathcal{A}}, \mathcal{S}_{\mathcal{E}})$, where $\mathcal{S}_{\mathcal{E}}$ is the set of directed edges. The directed graph regulates the executed sequence of agents and inter-agent information transmission pathways, varying different multi-agent structures. Based on the aforementioned motivation, we introduce an external dual latent memory system $\mathcal{M}$ attached to the VMAS, which is implemented in a model-agnostic manner with all base agents kept frozen, preserving the generalization and efficiency. As shown in Figure 4, this system comprises two core procedures, synthesis and orchestration of the dual latent memory, respectively.

In **memory synthesis** (Section 3.2), the preceding agents dynamically establish and update the dual latent memory system, as formulated below:

$$\{\mathcal{S}_1, \mathcal{O}_1 \ldots, \mathcal{S}_{n-1}, \mathcal{O}_{n-1}\} \to \mathcal{M}, \tag{1}$$

where $\mathcal{S}_i$ denotes the operational state, including the system prompt, accumulated inter-agent transmissive contexts from predecessor agents, and the historical status of the current agent, and $\mathcal{O}_i$ stands for the output of the agent.

In **memory orchestration** (Section 3.3), the current agent queries task-relevant contextual information for the memory system on demand, to seamlessly inform its perception and thinking during the inference generation process:

$$\{\mathcal{A}_n, \mathcal{M}, \mathcal{S}_n, \mathcal{T}_n\} \to \mathcal{O}_n, \tag{2}$$

where $\mathcal{T}_n$ represents the assigned task, including the textual task query and visual inputs.

## 3.2. Memory Synthesis

According to the propositional analyses, an ideal memory paradigm for VMAS entails that all preceding agents contribute to the update of system-shared memories. Thus, we separately initialize a latent perception memory $\mathcal{M}^P$ and a latent thinking memory $\mathcal{M}^T$, which encode the trajectories of preceding agents into structured memory units. Each memory unit $\mathbf{u}^{P/T}$ is persisted in a key-value pairs $(\mathbf{K}, \mathbf{V}^{P/T})$: the key $\mathbf{K}$ corresponds to a concise informational synopsis, while the value $\mathbf{V}^{P/T}$ stores the latent perception or thinking memory contents. For both of these two memories, we deploy a learnable compression module to distill the value of latent memory, thereby generating the universal key for retrieval:

$$\mathbf{K} = \mathcal{C}\left(\mathbf{V}^{P/T}\right), \tag{3}$$

where $\mathbf{K} \in \mathbb{R}^{d_{model}}$, and $\mathbf{V}^{P/T} \in \mathbb{R}^{l \times d_{model}}$. Here, $d_{model}$ denotes the hidden dimension of the model, and $l$ is the dynamic length. The compressor $\mathcal{C}\left(\cdot\right)$ is implemented through a lightweight transformer with masked attention (details are elaborated in Appendix C.1).

**Latent Perception Memory.** To provide more reliable and richer visual information, we incorporate multi-granularity perceptual observations into $\mathbf{V}^P$. Specifically, beyond the native visual inputs, we perform hierarchical downsampling to achieve a fine-to-coarse perception that spans from local details to global contexts. The native visual inputs are first partitioned into patch blocks, each with a size of $2^g$ times the original VLM vision patch size, where $g$ denotes the number of preset granularity levels. Then, for each patch block $\mathbf{X}_0$, we compute multi-granularity vision as follows:

$$[\mathbf{X}_0, \ldots, \mathbf{X}_{g-1}] = \{down_i\left(\mathbf{X}_0\right)\}_{i=0}^{g-1}, \tag{4}$$

where $down_i\left(\cdot\right)$ denotes the bilinear interpolation operation that downsamples the input to $1/2^i$ of the original spatial size of $\mathbf{X}_0$. To mitigate potential compatibility issues, we leverage the native vision encoder built into the visual agent and feed these multi-granularity patch features sequentially:

$$\left\{\mathbf{h}_1^{\mathbf{X}_i}, \ldots, \mathbf{h}_{l_i}^{\mathbf{X}_i}\right\}_{i=0}^{g-1} = \{\phi\left(\mathbf{X}_i\right)\}_{i=0}^{g-1}, \tag{5}$$

where $\phi\left(\cdot\right)$ is the visual encoding and alignment, $\mathbf{h}^{\mathbf{X}_i}$ denotes the hidden state sequence extracted from $\mathbf{X}_i$, and the corresponding sequence length is given by $l_i = 2^{g-2i+1}$ (details are shown in Appendix C.2). Empirical results confirm that $g$ is set to 3, corresponding to local, regional, and global granularity levels, balancing performance and computational overhead. After concatenation, the projected sequences of different granularity levels are performed as the value $\mathbf{V}^P$ of the perception memory unit $\mathbf{u}^P$.

Subsequently, we compute the key component $\mathbf{K}$ corresponding to each derived value component, thereby constructing a complete perception unit $\mathbf{u}^P = \left(\mathbf{K}, \mathbf{V}^P\right)$ and integrating it into the perception memory $\mathcal{M}^P$. Considering that visual encoding is computationally intensive, we initialize the latent perception memory before the first agent and expand it when new visual content is input.

**Latent Thinking Memory.** The original thinking trajectory is a continuous sequence of hidden states; storing it as a whole would lead to semantic confusion and retrieval difficulties. Empirical findings (Yu et al., 2026; Hu et al., 2026; Yin et al., 2025; Dong et al., 2025) demonstrate that language models exhibit higher uncertainty and entropy increase in the semantic boundaries; besides, the study (Chen et al., 2025a; Zhang et al., 2025a) indicates that intervening between sentences is more efficient to guide the thinking paths. Therefore, we decompose the thinking trajectory into semantically independent chunks at delimiter positions marked by high entropy, with each chunk corresponding to a complete and independent semantic unit. We first define a delimiter token set $\mathcal{S}_d$, and determine whether to truncate only when the current token falls in $\mathcal{S}_d$. The decision

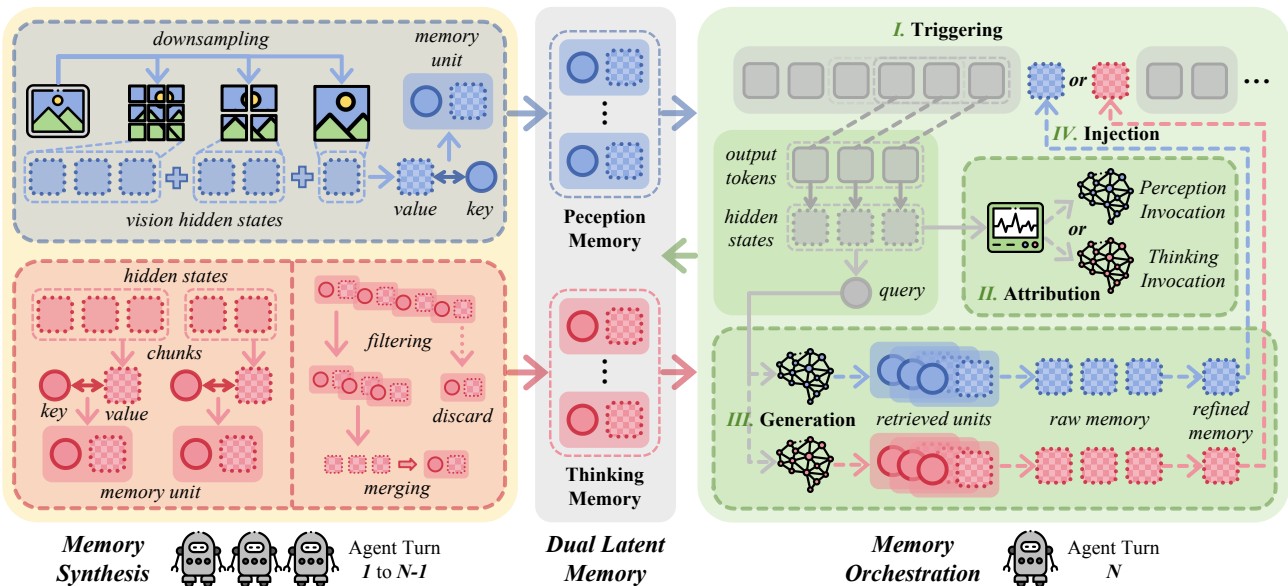

Figure 4. The overview of our proposed L²-VMAS.

process could be summarized as:

$$\mathrm{B}(\pi_i), \quad \pi_i = \begin{cases} 0, & t_i \notin S_d, \\ \mathrm{clip}\left(\dfrac{\mathbf{H}_i}{\log|\mathcal{V}|}, 0, 1\right), & t_i \in S_d, \end{cases} \quad (6)$$

where $\mathbf{H}_i$ represents the entropy of token generation at step $i$ (details in Appendix C.3), and $\mathbf{t}_i$ denotes current predicted token. Since entropy is bounded by $[0, \log|\mathcal{V}|]$, we perform normalization to obtain a Bernoulli probability $\mathrm{B}(\cdot)$, and apply clipping for stability, where $|\mathcal{V}|$ is the vocabulary size. It ensures that chunk boundaries are semantically significant at minimal computational cost. We treat the thinking segment between each two adjacent truncation decision $i, j$ as a chunk, and extract the hidden state sequence $\{\mathbf{h}_i^T, \dots, \mathbf{h}_j^T\}$ as the value $\mathbf{V}^T$ of the thinking memory unit $\mathbf{u}^T$.

Unlike the perception memory, we impose a predefined maximum capacity constraint $N$ on the latent thinking memory bank, to avoid unbounded memory expansion and enhance the scalability of agent turns. When there is no capacity overflow, the decomposed cognitive chunks are directly appended to $\mathcal{M}^T$; upon overflow, a dynamic memory management mechanism is activated to prune redundant information. To this end, we first define triggering rate $\mathbf{r}$ and global semantic similarity $\mathbf{s}$. For a thinking memory unit $\mathbf{u}^T$, the triggering rate $\mathbf{r}$ refers to the ratio of the cumulative triggering count of $\mathbf{u}^T$ to the total count of the entire memory system since the unit is incorporated into the memory bank. The global semantic similarity $\mathbf{s}$ is calculated as follows:

$$\mathbf{s} = \frac{1}{m-1} \sum_{t=1, t \neq i}^{m} \mathrm{sim}\left(\mathbf{K}_i^T, \mathbf{K}_t^T\right), \quad (6)$$

where $i$ denotes the index of the target thinking memory unit, and $m$ represents the current number of units residing in the thinking memory bank.

We then partition all memory units into five equal-sized quantiles, ordered by their triggering rate $\mathbf{r}$, denoted sequentially as Set 1 to Set 5. The dynamic management of memory is outlined in the following steps: First, we filter out units in Set 5 with weak semantic relevance, specifically those whose $\mathbf{s}$ values fall below the average of the entire memory bank, thereby releasing the invalid memory space. Subsequently, we designate units in Set 1 as merging bases $\mathbf{V}_{base}^T$. For each unit in Set 4 whose $\mathbf{s}$ is below the average, we identify its most semantically similar counterpart within the merging base set. We then fuse the value content of the base unit and the selected similar units, as below:

$$\mathbf{V}_{new}^T = \mathcal{C}_{merge}\left(\mathrm{concat}\left(\left[\mathbf{V}_{base}^T; \{\mathbf{V}_i^T\}_{i=1}^n\right]\right)\right) \quad (7)$$

where $\mathcal{C}_{merge}(\cdot)$ denotes the merging compressor that employs a homologous architecture with $\mathcal{C}(\cdot)$ to ensure the length of the updated value is not inflationary, and $n$ denotes the number of selected similar units. Finally, we update the key $\mathbf{K}$ corresponding to the merged base unit and reinsert it into the thinking memory bank.

### 3.3. Memory Orchestration

We propose a proactive memory invocation mechanism that activates memory resources only on demand and enforces the decoupled activation of perception and thinking memories. This mechanism unfolds via a four-stage sequential workflow: triggering, attribution, generation, and injection. Inspired by the insights from (Wang et al., 2025d), which

demonstrate that sustained high-entropy regions correspond to decision bifurcation points, signaling elevated uncertainty, we introduce an adaptive window strategy during autoregressive decoding to determine the timing of latent memory triggering. In contrast to semantic boundary truncation (as in Equation 6), memory can be triggered at arbitrary positions within the generation, a human-like cognitive paradigm in which memory retrieval occurs dynamically throughout the thinking and perception processes. We first calculate the average value $\mu_i$ and standard deviation $\sigma_i$ of the entropy sequence between $i - W + 1$ to $i$, where $W$ denotes the window length. The memory trigger decision at the $ith$ step:

$$\text{Trigger}(i) \Longleftrightarrow \left(\bar{H}_i > \mu_i + \lambda\sigma_i\right) \wedge \left(i - i_{\text{last}} \geq W\right), \quad (8)$$

where $\text{Trigger}(\cdot)$ denotes a boolean indicator, $\bar{H}_i$ stands for the average entropy between $i - 2W + 1$ to $i$ for smoother and stable triggering, $\lambda$ represents a threshold scaling factor that regulates the sensitivity. The final judgment term is a constraint that two continuous triggers within a length $W$ are forbidden to avoid frequent invocation.

Upon triggering, we extract the hidden state sequence $\left\{\mathbf{h}_{i-W+1}^T, \ldots, \mathbf{h}_i^T\right\}$ within the sliding window of length $W$, where the sequence serves as attribution hints and query, respectively. To determine whether to inject the perception memory or the thinking memory, we introduce a learnable gate $\sigma(\cdot)$, employing a Gumbel-Sigmoid function with temperature annealing to smooth discrete binary decisions and mitigate gradient vanishing issues during training (details in Appendix C.4). Similar to the acquisition of the key of the memory unit, we reuse the compression module $\mathcal{C}$ to distill a compact query. Subsequently, we retrieve similarity-based top-k memory units $\left\{\mathbf{u}_i^{P/T}\right\}_{i=1}^{k}$ from the target memory bank (either $\mathcal{M}^P$ or $\mathcal{M}^T$), and concatenate the values of retrieved memory units as raw memory content. To ensure the consistency, we refine the raw memory content as follows:

$$\mathbf{M}^{P/T} = \mathcal{C}_{refine}\left(\text{concat}\left(\left\{\mathbf{V}_i^{P/T}\right\}_{i=1}^{k}\right)\right), \quad (9)$$

where $M^{P/T} \in \mathbb{R}^{L \times d_{model}}$ denotes the final latent memory to be injected, $L$ is the preset memory sequence length, and $\mathcal{C}_{refine}(\cdot)$ denotes a memory refinement compressor. Finally, the latent memory $\mathbf{M}^{P/T}$ is seamlessly inserted at the triggering position, and the autoregressive decoding process resumes after memory injection.

### 3.4. Training Recipe

Throughout the training process, we keep the base VLM backbone of each agent frozen while solely training the custom-designed external memory components. We adopt a three-stage RL-driven training scheme: (1) Stage I: The latent memory is activated randomly to optimize memory construction and update mechanisms. (2) Stage II: The memory synthesis components are frozen, and the memory orchestration components are trained in isolation to improve memory recall ability. (3) Stage III: All external memory components are unlocked for joint training to optimize end-to-end performance. More training details and parameter configurations are in Appendix C.5.

## 4. Experiments

### 4.1. Settings

**Baselines.** To demonstrate the superiority of our dual latent memory, we compare it against both the single- and multi-agent baselines. The former uses a single agent to perform standard autoregressive decoding, while the latter constructs a VMAS with conventional inter-agent text-based information transmission. We select five widely adopted VLMs as the backbone of the agent: GLM-4.1V-Thinking (Hong et al., 2025), InternVL-3.5-8B (Wang et al., 2025e), LLaVA-OV-1.5-8B (An et al., 2025), and Qwen3-VL-8B-Thinking/Instruct (Bai et al., 2025). Additionally, we cover four model sizes, and six multi-agent structures: linear, layered, centralized, random, complete, and dynamic, as described in Appendix D.1.

**Benchmarks.** Our evaluations include four comprehensive visual benchmarks, including: MMbench (Liu et al., 2024), MMStar (Chen et al., 2024), RealWorldQA (xAI, 2024), and SimpleVQA (Cheng et al., 2025), assessing the visual perception, thinking, and advanced abilities. Additionally, we include four multi-image and video-based benchmarks: MuirBench (Wang et al., 2025b), BLINK (Fu et al., 2024), MVBench (Li et al., 2024), and LVBench (Wang et al., 2025f), for advanced visual scenarios assessment.

**Implementation.** During the three-stage training process, we leverage the large-scale GQA dataset (Hudson & Manning, 2019) to cultivate memory abilities, with no exposure to the test benchmarks. All experiments are conducted on 8 NVIDIA H200 141G GPUs. The window length $W$ and threshold scaling factor $\lambda$ are set to 16 and 0.5, for moderate memory triggering. The memory length $L$ and the maximum capacity of the thinking memory $N$ are set to 8 and 50, respectively. More details are in Appendix C.5 and D.1.

### 4.2. Main Results

**Performance Improvements to VMAS.** As presented in Table 1, existing VMAS yield marginal accuracy gains over their single-model counterparts, yet incur an exponential increase in token overhead. This leads to a suboptimal accuracy-cost trade-off and thus offers limited practical utility in real-world deployment scenarios. In contrast, our $L^2$-VMAS framework delivers average accuracy improvements of 2.7-5.4% across all five backbones, accompanied

*Table 1.* Results of five base models based on dynamic structure, comparing both accuracy and total token usage. The best values are **bolded**, the rightmost column shows the average results, and the arrows indicate the improvement or reduction compared to VMAS.

| Backbone | Method | MMBench | | MMStar | | RealWorldQA | | SimpleVQA | | Average | |
|---|---|---|---|---|---|---|---|---|---|---|---|
| | | *Accuracy* | *Token* | *Accuracy* | *Token* | *Accuracy* | *Token* | *Accuracy* | *Token* | *Accuracy* | *Token* |
| GLM-4.1V-9B-Thinking | Single | 80.4 | 482 | 71.5 | 649 | 69.0 | 708 | 46.3 | 546 | 66.7 | 596 |
| | VMAS | 80.2 | 4416 | 71.8 | 6744 | 67.2 | 7446 | 47.5 | 5639 | 66.6 | 6061 |
| | L²-VMAS | **82.4** ↑2.2 | **2487** ↓1929 | **74.1** ↑2.3 | **3451** ↓3293 | **70.3** ↑3.1 | **4314** ↓3132 | **51.3** ↑3.8 | **3125** ↓2514 | **69.5** +4.3% | **3344** -44.8% |
| InternVL-3.5-8B | Single | 79.5 | 416 | 69.0 | 454 | 67.4 | 523 | 41.1 | 452 | 64.3 | 3461 |
| | VMAS | 81.7 | 2732 | 72.5 | 2958 | 72.7 | 3360 | 45.0 | 2993 | 68.0 | 3011 |
| | L²-VMAS | **82.2** ↑0.5 | **2024** ↓708 | **74.7** ↑2.2 | **2189** ↓769 | **74.8** ↑2.1 | **2534** ↓826 | **47.6** ↑2.6 | **2233** ↓760 | **69.8** +2.7% | **3011** -25.4% |
| LLaVA-OV-1.5-8B | Single | 82.6 | 340 | 67.0 | 388 | 68.3 | 445 | 44.6 | 389 | 65.6 | 391 |
| | VMAS | 84.7 | 2278 | 73.1 | 2552 | 72.8 | 2891 | 46.8 | 2567 | 69.4 | 2572 |
| | L²-VMAS | **84.4** ↓0.3 | **1824** ↓454 | **75.1** ↑2.0 | **2067** ↓485 | **76.2** ↑3.4 | **2135** ↓756 | **50.7** ↑3.9 | **2069** ↓498 | **71.6** +3.2% | **2024** -21.3% |
| Qwen3-VL-8B-Instruct | Single | 84.0 | 318 | 70.4 | 363 | 71.0 | 402 | 50.1 | 358 | 68.9 | 360 |
| | VMAS | 84.9 | 2190 | 74.8 | 2467 | 74.7 | 2769 | 51.4 | 2492 | 71.4 | 2480 |
| | L²-VMAS | **87.4** ↑2.5 | **1682** ↓508 | **77.5** ↑2.7 | **1907** ↓560 | **77.6** ↑2.9 | **2101** ↓668 | **53.8** ↑2.4 | **1865** ↓627 | **74.1** +3.7% | **1889** -23.8% |
| Qwen3-VL-8B-Thinking | Single | 84.8 | 557 | 75.5 | 687 | 72.9 | 730 | 49.4 | 604 | 70.9 | 645 |
| | VMAS | 85.3 | 5241 | 78.1 | 6670 | 76.2 | 7026 | 50.5 | 5679 | 72.5 | 6154 |
| | L²-VMAS | **88.8** ↑3.5 | **2983** ↓2258 | **81.4** ↑3.3 | **3677** ↓2993 | **80.2** ↑4.0 | **3976** ↓3050 | **55.3** ↑4.8 | **3256** ↓2423 | **76.5** +5.4% | **3473** -43.6% |

*Table 2.* Results of various model size based on dynamic structure and Qwen3-VL family.

| Backbone | Method | MMBench | | MMStar | | RealWorldQA | | SimpleVQA | | Average | |
|---|---|---|---|---|---|---|---|---|---|---|---|
| | | Instruct | Thinking | Instruct | Thinking | Instruct | Thinking | Instruct | Thinking | Instruct | Thinking |
| Qwen3-VL-2B | Single | 79.0 | 79.3 | 57.8 | 69.1 | 63.7 | 68.6 | 40.5 | 42.6 | 60.3 | 64.9 |
| | VMAS | 84.6 | 84.2 | 65.0 | 72.5 | 68.8 | 72.6 | 47.9 | 47.4 | 66.6 | 69.2 |
| | L²-VMAS | 84.2 ↓0.4 | **85.8** ↑1.6 | 66.8 ↑1.8 | **73.6** ↑1.1 | 68.6 ↓0.2 | **73.7** ↑1.1 | 50.0 ↑2.1 | 49.6 ↑2.2 | 67.4 ↑1.3% | 70.7 ↑2.2% |
| Qwen3-VL-4B | Single | 83.4 | 84.1 | 69.4 | 74.3 | 70.5 | 72.5 | 48.2 | 48.0 | 67.9 | 69.7 |
| | VMAS | 85.7 | 85.4 | 73.9 | 75.3 | 75.9 | 75.3 | 50.4 | 49.5 | 71.5 | 71.4 |
| | L²-VMAS | 87.2 ↑1.5 | **88.6** ↑3.2 | 76.4 ↑2.5 | **77.1** ↑1.8 | 77.4 ↑1.5 | **77.1** ↑1.8 | 53.8 ↑3.4 | 53.3 ↑3.8 | 73.7 ↑3.1% | 74.0 ↑3.7% |
| Qwen3-VL-8B | Single | 84.0 | 84.8 | 70.4 | 75.5 | 71.0 | 72.9 | 50.1 | 49.4 | 68.9 | 70.9 |
| | VMAS | 84.9 | 85.3 | 74.8 | 78.1 | 74.7 | 76.2 | 51.4 | 50.5 | 71.4 | 72.5 |
| | L²-VMAS | 87.4 ↑2.5 | **88.8** ↑3.5 | 77.5 ↑2.7 | **81.4** ↑3.3 | 77.6 ↑2.9 | **80.2** ↑4.0 | 53.8 ↑2.4 | 55.3 ↑4.8 | 74.1 ↑3.7% | 76.5 ↑5.4% |
| Qwen3-VL-32B | Single | 87.2 | 89.0 | 76.8 | 79.7 | 79.1 | 78.4 | 56.7 | 55.3 | 75.0 | 75.6 |
| | VMAS | 89.1 | 87.3 | 77.8 | 77.6 | 78.0 | 77.6 | 56.8 | 57.8 | 75.4 | 75.1 |
| | L²-VMAS | **90.6** ↑1.5 | **90.0** ↑2.7 | **79.4** ↑1.6 | **80.2** ↑2.6 | **80.5** ↑2.5 | **80.3** ↑2.7 | **59.4** ↑2.6 | **60.6** ↑2.8 | **77.5** ↑2.7% | **77.8** ↑3.6% |

by a remarkable reduction in total token consumption of 21.3-44.8%. Notably, our method achieves more substantial performance gains in complex tasks and scenarios that demand intensive inter-agent information transmission. For instance, it achieves a 5.4% performance boost and a 43.6% reduction in tokens on Qwen3-VL-8B-Thinking, outperforming the gains on Instruct model by a distinct margin. From a benchmarking perspective, it achieves significantly more pronounced improvements on RealWorldQA and SimpleVQA. Additionally, as shown in Table 5, our method still exhibits significant improvements over the baselines in augmented visual scenarios, including multi-image and video-based tasks. It yields an average performance enhancement of over 4%, underscoring its robustness against data distribution shifts in enhanced visual scenarios.

**Robustness on Model Size and Multi-agent Structure.** As demonstrated in Table 2, L²-VMAS delivers consistent improvements over baselines across models spanning 2B/4B/8B/32B, indicating independence on a particular

model size. On the four-benchmark average, our method improves over VMAS by 1.3-3.7% in the Instruct model and by 2.2-5.4% in the Thinking model. Notably, gains are systematically larger in the Thinking model, suggesting that our proposed method more effectively amplifies the benefits of deep, sustained reasoning. In addition, Table 3 further examines performance across six multi-agent topologies: L²-VMAS consistently outperforms the corresponding baseline under each structure, indicating that the method does not require careful topology tuning to be effective. Averaged across benchmarks, it yields stable gains of 2.7-6.3% across structures. These experiments showcase that our proposed approach does not rely on coincidental model-, size-, or structure-specific adaptations, but rather enhances the general memory ability in VMAS.

### 4.3. Additional Analyses

**Effectiveness of Dual Memory System.** To further validate its superiority, we conduct a study on the MMStar bench-

*Table 3.* Results of various multi-agent structures based on Qwen3-VL-8B-Instruct/Thinking.

| Structure | Method | MMBench | | MMStar | | RealWorldQA | | SimpleVQA | | Average | |
|---|---|---|---|---|---|---|---|---|---|---|---|
| | | Instruct | Thinking | Instruct | Thinking | Instruct | Thinking | Instruct | Thinking | Instruct | Thinking |
| Linear | VMAS | 83.1 | 82.8 | 72.6 | 74.2 | 72.9 | 73.0 | 48.9 | 49.5 | 69.3 | 69.9 |
| | L²-VMAS | **84.5** ↑1.4 | **85.9** ↑3.1 | **76.2** ↑3.6 | **78.0** ↑3.8 | **73.1** ↑0.2 | **73.4** ↑0.4 | **51.1** ↑2.2 | **52.7** ↑3.2 | **71.2** ↑2.7% | **72.5** ↑3.7% |
| Layered | VMAS | 84.3 | 84.8 | 73.1 | 75.1 | 73.4 | 73.4 | 49.7 | 50.6 | 70.1 | 71.0 |
| | L²-VMAS | **85.7** ↑1.4 | **87.4** ↑2.6 | **76.4** ↑3.3 | **78.6** ↑3.5 | **74.8** ↑1.4 | **74.9** ↑1.5 | **51.6** ↑1.9 | **53.8** ↑3.2 | **72.1** ↑2.8% | **73.6** ↑3.8% |
| Centralized | VMAS | 84.2 | 84.0 | 73.8 | 74.9 | 72.7 | 72.6 | 48.6 | 49.0 | 69.8 | 70.1 |
| | L²-VMAS | **86.5** ↑2.3 | **87.3** ↑3.3 | **76.9** ↑3.1 | **79.4** ↑4.5 | **76.3** ↑3.6 | **77.2** ↑4.6 | **52.2** ↑3.6 | **54.3** ↑5.3 | **73.0** ↑4.5% | **74.6** ↑6.3% |
| Random | VMAS | 84.9 | 84.6 | 75.1 | 77.7 | 74.8 | 75.8 | 51.7 | 50.8 | 71.6 | 72.2 |
| | L²-VMAS | **86.9** ↑2.0 | **87.6** ↑3.0 | **77.8** ↑2.7 | **81.3** ↑3.6 | **77.3** ↑2.5 | **80.0** ↑4.2 | **53.5** ↑1.8 | **54.8** ↑4.0 | **73.9** ↑3.1% | **75.9** ↑5.1% |
| Complete | VMAS | 84.8 | 83.9 | 75.3 | 77.4 | 75.3 | 75.0 | 52.0 | 50.6 | 71.9 | 71.7 |
| | L²-VMAS | **87.2** ↑2.4 | **87.2** ↑3.3 | **77.6** ↑2.3 | **80.9** ↑3.5 | **77.6** ↑2.3 | **79.7** ↑4.7 | **54.1** ↑2.1 | **55.0** ↑4.4 | **74.1** ↑3.1% | **75.7** ↑5.5% |
| Dynamic | VMAS | 84.9 | 85.3 | 74.8 | 78.1 | 74.7 | 76.2 | 51.4 | 50.5 | 71.4 | 72.5 |
| | L²-VMAS | **87.4** ↑2.5 | **88.8** ↑3.5 | **77.5** ↑2.7 | **81.4** ↑3.3 | **77.6** ↑2.9 | **80.2** ↑4.0 | **53.8** ↑2.4 | **55.3** ↑4.8 | **74.1** ↑3.7% | **76.5** ↑5.4% |

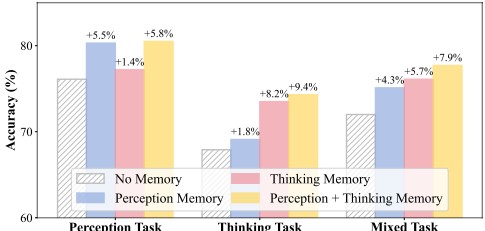

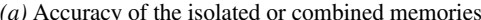

*(a)* Accuracy of the isolated or combined memories.

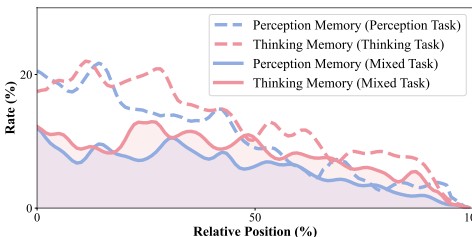

*(b)* Triggering rate of the two memories.

*Figure 5.* Effectiveness Analyses of the dual memory system based on perception, thinking, and mixed tasks.

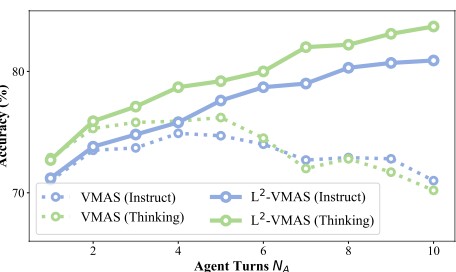

*Figure 6.* Comparison of scalability between VMAS and L²-VMAS based on RealWorldQA benchmark.

**Scalability.** As revealed in Figure 2 and 7, the baselines undergo performance degradation, which starts to decline when turns rise to only 3, and by turn 10, it drops even below the single-agent level. Conversely, compared with VMAS, our method achieves steady performance gains of 13.9% and 19.2% for the Instruct and Thinking models, respectively, as agent turns increase. At the 10*th* turn, our method improves by 13.6% and 15.1% over the 1*st* turn of the two backbones, enabling more turns and greater scalability.

**Ablation & Sensitivity & Generalization Analyses.** We perform ablation as shown in Table 6, demonstrating the necessity of each component in L²-VMAS, including details on memory synthesis and orchestration. The sensitivity analyses in Tables 7, 8, and 9 quantify the impact of hyperparameters on the balance between efficiency and effectiveness, and the additional improvements as the parameter settings scale further. In addition, as reported in Table 10, which evaluates models on unseen benchmarks, it exhibit greater generalization ability than the baselines.

## 5. Conclusion

In this work, we quantitatively identify the problem of "scaling wall" in conventional VMAS, *i.e.*, deeper agent turns counter-intuitively leads to worse performance and undesirable computation cost. We trace the root cause to the

mark (Chen et al., 2024), evaluating performance across three task subsets: perception, thinking, and mixed tasks. As illustrated in Figure 5a, both the perception memory and thinking memory consistently yield performance gains over the raw VMAS without memory. Notably, each memory type delivers more pronounced improvements on its corresponding task category: perception memory boosts accuracy by 5.5% on perception tasks (versus 1.4%), while thinking memory elevates accuracy by 8.2% on thinking tasks (versus 1.8%). Furthermore, in mixed tasks, the combination of memories achieves the highest accuracy gain of 7.9%, underscoring the value of decoupled but complementary memories. As depicted in Figure 5b and 10, the curves of triggering rate and relative position of the two memories demonstrate that the triggering is self-adaptive, adjusting dynamically according to various tasks.

information bottleneck inherent in text-centric communication, which fails to preserve visual perception and cognitive thinking simultaneously across agents. To this end, we propose $L^2$-VMAS, which designs a dual latent memory framework decoupling perception and thinking, respectively. In addition, we introduce an entropy-driven proactive triggering and retrieval mechanism. Extensive experiments confirm the efficacy of our method, which we believe offers a scalable path forward for developing more reliable and complicated VMAS for future research.

## Impact Statement

This paper presents work whose goal is to advance the field of machine learning. There are many potential societal consequences of our work, none of which we feel must be specifically highlighted here.

## Acknowledgment

This work was supported by the National Natural Science Foundation of China under Grant No. 62320106007, and by NUS Grant A-0010106-00-00, A-8004365-00-00, and A-8004410-01-00.

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

# Appendix

# A. Related Works

### A.1. Visual Multi-agent System

Existing research on VMAS primarily centers on collaborative perception and cognition among visual agents and the mechanisms governing inter-agent collaboration. Along one research vein, efforts focus on designing task-adaptive multi-agent frameworks tailored to visual scenarios (Wang et al., 2023; Elhenawy et al., 2024; Huang et al., 2025a). A complementary line of work investigates foundational components such as training paradigms (Zhang et al., 2022; 2025c), sequential planning strategies (Brienza et al., 2024), and theoretical guarantees (WU et al., 2025; Shi et al., 2025), alongside the development of specialized VMAS benchmarks (Jiang et al., 2024; Du et al., 2023). Additionally, VMAS has been extended to diverse downstream applications, including structured visual understanding (*e.g.*, document and chart) (Han et al., 2025; Yu et al., 2025a; Li et al., 2025b; Zhang et al., 2025d), medical analysis (Yang et al., 2025a; Feng et al., 2025b), and embodied intelligence (Liu et al., 2022; Kang et al., 2025a; Hanlon et al., 2024).

Despite these advancements, the vast majority of existing VMAS paradigms remain anchored to text-centric inter-agent communication pipelines, succumbing to the critical limitations outlined in Section 3.1. While recent works have explored latent-space communication paradigms to bypass natural language bottlenecks (Zheng et al., 2025; Fu et al., 2025; Zou et al., 2025), these approaches are not directly transferable to VMAS. Their design fails to account for the unique challenges posed by high-dimensional visual inputs and the inherent risk of perceptual-cognitive information conflation—core pain points that motivate our decoupled dual memory architecture.

### A.2. Visual Latent Space

As a machine-native, high-density representational medium, the visual latent space enables direct interoperability with the latent states of VLMs (Yu et al., 2026). Building on this inherent advantage, three distinct research strands have emerged in the realm of latent visual representation: One prominent line of work enables models to perform reasoning directly over visual latent tokens rather than textual reasoning tokens, thereby eschewing the inefficiencies of text-centric inference pipelines. Representative approaches include LaCoT (Sun et al., 2025), CoVT (Qin et al., 2025), MCOUT (Pham & Ngo, 2025), DMLR (Liu et al., 2025), Monet (Wang et al., 2025c), and LaViT (Wu et al., 2026). Another complementary paradigm leverages external models or annotated visual datasets to internalize latent visual representations into VLMs, then inserts these latent visual tokens into token sequences to provide explicit visual observations. Typical methods in this category include Mirage (Yang et al., 2025b), LVR (Li et al., 2025a), 3DThinker (Chen et al., 2025b), and Latent Sketchpad (Zhang et al., 2025b). A third research strand focuses on latent visual memory modules that store reusable visual experience to enhance the long-term visual performance, with notable works including CoMEM (Wu et al., 2025a), CoMEM-Agent (Wu et al., 2025b), and VisMem (Yu et al., 2025b). However, none of these paradigms can be directly applied to VMAS because they do not address the core problems of efficiency, proactivity, and scalability.

# B. Requisite Analyses

### B.1. Experimental Details

**Accuracy and Token Usage Across Agent Turns.** The range of agents is 1 to 10, with each representing the complete operation procedure of a single agent. We calculate both task accuracy and total token usage. The former metric quantifies performance as the percentage of tasks correctly answered. The latter reflects the integrated computational and communicative efficiency, calculated as the sum of tokens used and generated by each agent per turn, including prompt tokens, visual feature tokens, instruction tokens (including information transmitted from previous agents), and output tokens.

**Inter-agent Transmission Content.** The four groups in this experiment are: (a) Conclusion-only: Solely transmits the final task conclusion of each agent, while omitting all intermediate perceptual observations and thinking steps, and auxiliary annotations; (b) Perception: Transmits visual perceptual observations alongside the conclusion; (c) Thinking: Transmits thinking trajectories alongside the conclusion; (d) Full-content: Transmits the complete unfiltered output of each agent, integrating unstructured perception, thinking trajectories, and the conclusion, consistent with the default communication paradigm of existing VMAS. Here, we use GPT-5.1 (OpenAI, 2025) to extract perception and thinking contents structurally, and the prompt we use is as follows:

---

**Prompt for Evaluating the Severity of Hallucination $h$**

```
You are a strict information extraction engine.
You MUST only copy spans from the input text verbatim.
Do NOT paraphrase, do NOT add new content, do NOT explain reasoning.
Output MUST be valid JSON only.

Task: Extract PERCEPTION, THINKING, and CONCLUSION segments from the raw agent output.

Definitions:
- PERCEPTION: directly observable visual facts (texts seen, positions, colors, UI elements, state changes).
    No inference.
- THINKING: reasoning, planning, hypothesis, strategy, self-reflection. Not raw observation.
- CONCLUSION: the final decision/answer/result of the task.

Return JSON with:
perception: list of {text,start,end}
thinking: list of {text,start,end}
conclusion: list of {text,start,end}
notes: {ambiguous:boolean, reason:string}

Raw text:
```

## B.2. Additional Results

In order to further verify our obtained insights (see in Section 2.2) and ensure the consistency of the results on different data distributions, we conduct the two experiments on four benchmarks, *i.e.*, MMbench (Liu et al., 2024), MMStar (Chen et al., 2024), RealWorldQA (xAI, 2024), and SimpleVQA (Cheng et al., 2025). As shown in Figure 7 and 8, for all results on the four benchmarks, we can reach the same conclusion in terms of the accuracy and total token usage among agents, as well as the comparison of inter-agent information transmission.

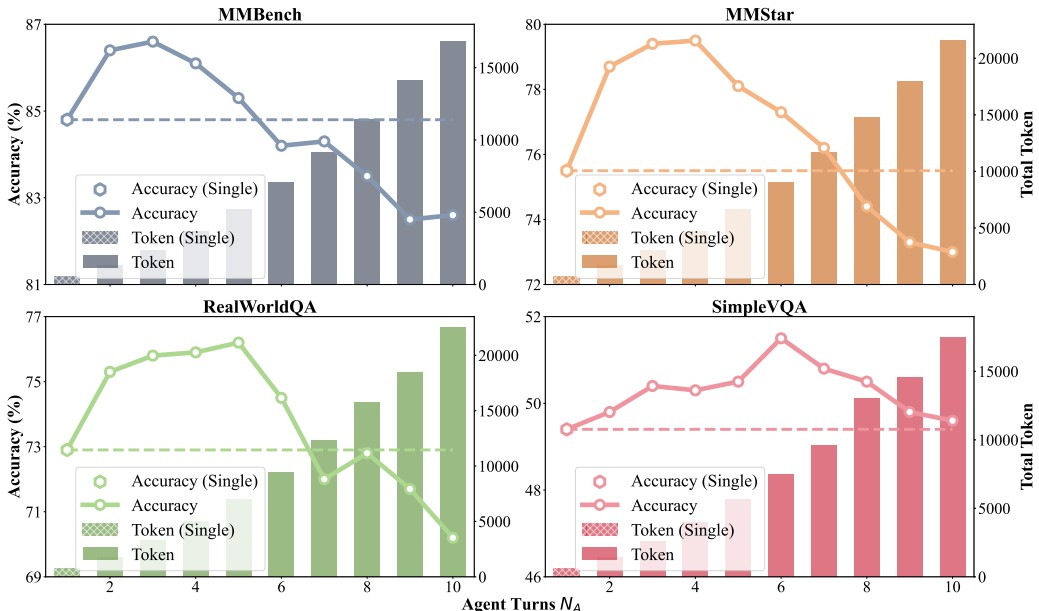

*Figure 7.* Accuracy and total token usage among agent turns on four benchmarks.

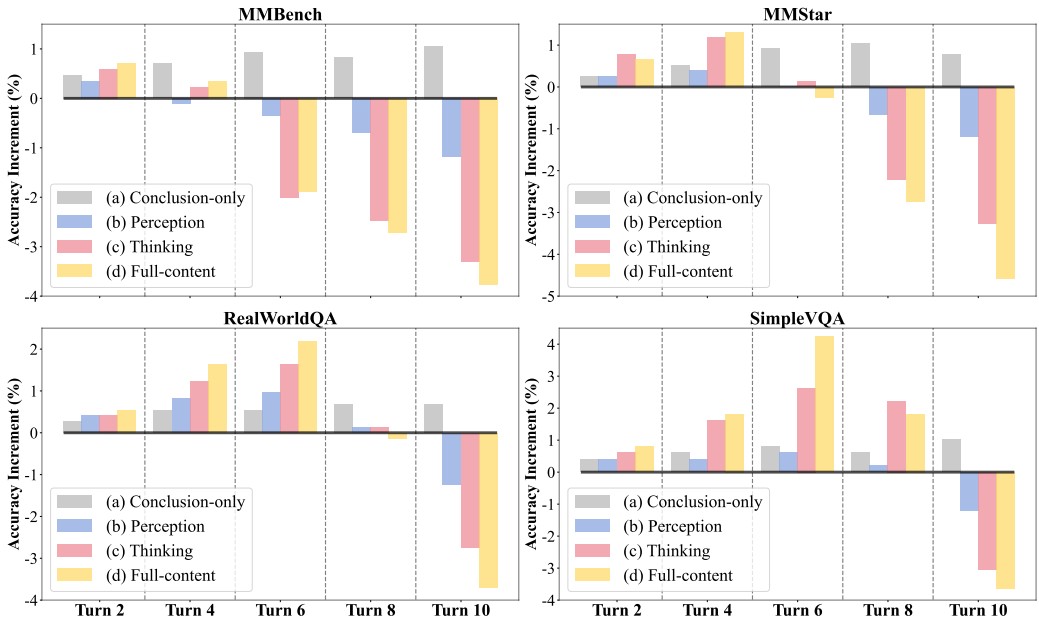

*Figure 8.* Comparison of inter-agent transmission content on four benchmarks.

## C. Methodologies

### C.1. Compression Module

We implement the three compression modules $\mathcal{C}$ (see in Equation 3), $\mathcal{C}_{merge}$ (see in Equation 7), and $\mathcal{C}_{refine}$ (see in Equation 9) as lightweight transformer encoders that compresses the content sequence into a fixed-length sequence. Here, we perform unified modeling on these three compressors with identical structures. Given the sequence to be compressed $S_{compress} \in \mathbb{R}^{x \times d_{model}}$, we first append a learnable token sequence $S_{target} \in \mathbb{R}^{y \times d_{model}}$ to form an input sequence $z = [S_{compress}; S_{target}]$. This sequence is then fed into a multi-layer transformer. Each layer $\ell$ of the compressor performs masked self-attention followed by a feed-forward update. Formally, for input $z^{\ell-1}$ at layer $\ell$, the self-attention output is computed as:

$$\text{SA}_{\mathcal{C}}(z^{\ell-1}) = \text{SM}\left(\frac{(z^{\ell-1}W_q)(z^{\ell-1}W_k)^{\top}}{\sqrt{d_k}} + M_c\right)(z^{\ell-1}W_v), \tag{10}$$

$$z^{\ell} = \text{FF}\left(\text{LN}\left(z^{\ell-1} + \text{SA}_{\mathcal{C}}\left(\text{LN}(z^{\ell-1})\right)\right)\right) + z^{\ell-1}, \tag{11}$$

where $\text{SM}(\cdot)$ denotes the softmax operation, and $M_c$ is the attention mask matrix for the compressor. We then apply a residual connection with layer normalization (LN) and a position-wise feed-forward network (FF) to obtain the output of layer $\ell$, as shown in the second line above. The mask $M_c$ is designed to allow the compression query token to attend to all memory content tokens, while preventing the memory tokens from attending to the query token. If we index the tokens in $z$ such that the first $x$ tokens correspond to $S_{compress}$ and the last $y$ token corresponds to $S_{target}$, then $M_c$ is defined as:

$$(M_c)_{ij} = \begin{cases} -C, & i \in \{1,\dots,x\}, \ j = \{x+1,\dots,x+y\} \\ 0, & \text{otherwise}, \end{cases} \tag{12}$$

where $C \gg 0$ is a large constant (so that adding $-C$ to those attention logits effectively masks out that attention). This masking strategy ensures that the compression token sequence $S_{target}$ can attend to the content in $S_{compress}$ to gather information; otherwise, it is not permitted, preventing information flow from the query back into the memory content. After processing through all layers, the output embedding corresponding to the compression token sequence is used as the compressed output.

## C.2. Visual Encoding and Alignment

As provided in Equation 5, $\phi\left(\cdot\right)$ is the process of visual encoding and alignment. Formally, let each vision encoder output $\mathbf{h}^{\mathbf{X}i}$ be the hidden state extracted from the $ith$ granularity input $\mathbf{X}_i$. To align the visual hidden states with the language model's space, we leverage the initial pre-trained projector rather than introducing a new alignment network. In different VLM families, the details of alignment structure vary. If $\mathbf{h}_t^{X_i} \in \mathbb{R}^{d_v}$ is a hidden state from the vision encoder (with $d_v$ the vision feature dimension) and $d_l$ is the embedding dimension of the language model, then the projector can be represented by a weight matrix $\mathbf{W}_p \in \mathbb{R}^{d_l \times d_v}$ and bias $\mathbf{b}_p \in \mathbb{R}^{d_l}$. The projector maps each visual hidden state to the language space as:

$$\tilde{\mathbf{h}}_t^{X_i} = \mathbf{W}_p, \mathbf{h}_t^{X_i} + \mathbf{b}_p, \tag{13}$$

yielding an aligned feature $\tilde{\mathbf{h}}_t^{X_i} \in \mathbb{R}^{d_l}$ for each visual token. By reusing the VLM's native projection layer (kept frozen during our training), we ensure compatibility with the language representations without adding any new trainable alignment parameters.

We apply the projection function to every hidden state at each granularity level obtained from the vision encoder. This produces a sequence of language-space features $\left\{\tilde{\mathbf{h}}_1^{\mathbf{X}_i}, \ldots, \tilde{\mathbf{h}}_{l_i}^{\mathbf{X}_i}\right\}_{i=0}^{g-1}$ for each granularity level $i = 0, 1, \ldots, g-1$. Finally, we concatenate the projected sequences from all $g$ levels to construct the aggregated perceptual memory value:

$$\mathbf{V}^P = \text{concat}\left(\tilde{\mathbf{h}}^{X_0}, \tilde{\mathbf{h}}^{X_1}, \ldots, \tilde{\mathbf{h}}^{X_{g-1}}\right), \tag{14}$$

which serves as the value component $\mathbf{V}^P$ of the perception memory unit $\mathbf{u}^P$. This means that $\mathbf{V}^P$ contains a fine-to-coarse fused representation of the visual input, aligned to the language space and ready to be attended by the language model's decoder. Using the VLM's native projector in this way ensures that the visual information is injected in a form the language model can natively understand, without any additional alignment training.

## C.3. Entropy

To find the semantic boundaries (see in Equation 6) and decide the triggering (see in Equation 8), we utilize the entropy as the metric, which could be calculated as below:

$$\mathbf{H}_i = -\sum_{j=1}^{|\mathcal{V}|} \mathbf{p}_{i,j} \log \mathbf{p}_{i,j}, \tag{15}$$

where $\mathcal{V}$ denotes the vocabulary, and $\mathbf{p}$ means the probability of generating the $jth$ word at time step $i$:

$$\mathbf{p}_i = \pi\left(\cdot \mid \mathbf{z}_{<t}, \mathcal{T}\right) = \text{Softmax}\left(\frac{\mathbf{z}_t}{\iota}\right), \tag{16}$$

where $\pi$ denotes the language model, $\mathbf{z}$ stands for the predicted logits, and $\iota$ represents the decoding temperature.

## C.4. Learnable Gate

We introduce a learnable gating router to decide, at step $t$, whether to inject perception memory or thinking memory. Let the hidden-state window at the decision point be $\{\mathbf{h}_{i-W+1}^T, \ldots, \mathbf{h}_i^T\}$. A gating network $g\left(\cdot\right)$ aggregates this window and outputs a scalar logit:

$$a_t = g_\phi\left(\mathbf{H}_t\right), \qquad p_t = \text{Sigmoid}(a_t), \tag{17}$$

where $p_t$ denotes the probability of selecting one memory type, and the alternative is selected with probability $1 - p_t$.

Since hard binary routing is non-differentiable, we adopt a Gumbel-Sigmoid relaxation during training to obtain near-discrete decisions while retaining end-to-end differentiability. Concretely, we sample uniform noise and construct logistic noise as:

$$u_t \sim \text{Uniform}(0, 1), \qquad \gamma_t = \log u_t - \log(1 - u_t). \tag{18}$$

We then compute the relaxed gate value:

$$\tilde{z}_t = \text{Sigmoid}\left(\frac{a_t + \gamma_t}{\tau}\right) \in (0, 1), \tag{19}$$

where the temperature $\tau > 0$ controls the sharpness of the decision: larger $\tau$ yields smoother and more stochastic routing, whereas smaller $\tau$ pushes $\tilde{z}_t$ toward a binary value, approximating a hard switch. At inference time, $\tilde{z}_t$ can be thresholded to obtain a deterministic binary decision.

To stabilize optimization early while gradually encouraging near-discrete routing later, we anneal the temperature:

$$\tau(e) = \max\left(\tau_{\min},\ \tau_0 \cdot \lambda^e\right), \qquad 0 < \lambda < 1, \tag{20}$$

with training step index $e$. This schedule provides smoother gradients and better exploration at the beginning of training, and progressively reduces stochasticity to sharpen routing decisions, thereby mitigating vanishing gradients typically associated with hard binary routing.

### C.5. Training Recipe

Throughout the training process, we keep the pre-trained VLM backbone of each agent frozen and do not update it, while only training the external latent-variable dual-memory modules. This approach thereby not only preserves generalization capability but also focuses learning capacity on memory synthesis and orchestration. We adopt a three-stage, RL-driven training, *i.e.*, proximal policy optimization (PPO) (Schulman et al., 2017), where the reward signal is primarily tied to the accuracy.

Stage I, which optimizes memory synthesis and updating capabilities to enable the system to learn how to encode the perception and thought trajectories of preceding agents into retrievable latent memories. During this stage, memory triggering occurs at random rather than relying on entropy-triggered mechanisms. This forces the system to undergo frequent memory writes and updates, capturing more contextual information and task states, thereby enhancing the robustness of memory encoding. In addition, we only unlock components of memory synthesis, *i.e.*, the compressor $\mathcal{C}\left(\cdot\right)$, and merging compressor $\mathcal{C}_{merge}\left(\cdot\right)$.

Stage II, which fixes memory synthesis, focuses on training to enable the system to proactively and on demand access memories. We freeze all memory synthesis components trained in Stage I to ensure the distribution of the memory repository remains stable and prevents training instability. In addition, we only unlocks the learnable gate $\sigma\left(\cdot\right)$, reused compressor $\mathcal{C}\left(\cdot\right)$, and refining compressor $\mathcal{C}_{refine}\left(\cdot\right)$.

Stage III, which unlocks all memory components to enable coordinated adaptation between synthesis and orchestration, further enhances overall end-to-end performance.

To obtain reproducible results, we provide all the parameters used in our research in the three-stage training recipe, as listed in Table 4.

## D. Experiments

### D.1. Settings

**Baselines.** We select two baselines for comparison, including a single agent and VMAS. The single-agent baseline uses a single instance of each VLM to perform independent autoregressive decoding. For each task, the input visual data (*e.g.*, images, video frames) is first encoded into visual tokens, then fed into the language decoder to generate the output sequentially along withe system and instruction tokens. The multi-agent baseline consists of several agents that collaborate via text-based communication. The collaboration workflow is as follows: First, each agent processes and generates outputs independently. These outputs are then systematically integrated as a core component of the input instruction for the subsequent agent in the topology chain. This sequential handover and iterative processing continue until the last agent in the sequence produces the final result, which constitutes the ultimate response to the original task.

**Multi-agent Structure.** Since our proposed method relies on the VMAS contexts, we first briefly delineate the six structures we used. As illustrated in Figure 9, we include six common structures: (1) linear, which implements a linear configuration for agent-mediated interactions; (2) layered, which comprises multiple hierarchical layers, where agent nodes within the current layer establish connections exclusively with those in the subsequent layer; (3) centralized, which exists as a central agent entity that is responsible for coordinating other agents, including the collection, transmission, and distribution of information. (4) random, which builds stochastic interconnections between agent nodes, where each agent dynamically redirects to the subsequent node based on contexts; (5) complete, adopts a fully-connected mesh topology, ensuring that each agent node can access any other node in the system via at least one viable path; (6) dynamic (Zhang et al., 2024), encodes

Table 4. Three-stage experimental details and configurations.

| Configuration | | Stage I | Stage II | Stage III |
|---|---|---|---|---|
| **Component** | VLM | - | - | - |
| | $\mathcal{C}(\cdot)$ | $\checkmark$ | $\checkmark$ | $\checkmark$ |
| | $\mathcal{C}_{merge}(\cdot)$ | $\checkmark$ | - | $\checkmark$ |
| | $\mathcal{C}_{refine}(\cdot)$ | - | $\checkmark$ | $\checkmark$ |
| | $\sigma(\cdot)$ | - | $\checkmark$ | $\checkmark$ |
| **Training Parameters** | Algorithm | PPO (Schulman et al., 2017) | | |
| | - clip_range $\epsilon$ | 0.2 | 0.2 | 0.1 |
| | - max_grad_norm | 0.5 | 0.5 | 0.5 |
| | - target_kl | 0.02 | 0.02 | 0.02 |
| | - gamma | 0.995 | 0.995 | 0.995 |
| | - gae_lambda | 0.95 | 0.95 | 0.98 |
| | steps | 100k | 80k | 50k |
| | learning_rate | $1 \times 10^{-4}$ | $5 \times 10^{-5}$ | $2 \times 10^{-5}$ |
| | lr_schedule | linear decay | | |
| | optimizer | Adam | | |
| | n_steps | 2048 | | |
| | num_envs | 8 | | |
| | mini_batch_size | 128/256 | | |
| **Hyper Parameters** | window length $W$ | 16 | | |
| | threshold scaling factor $\lambda$ | 0.5 | | |
| | memory length $L$ | 8 | | |
| | granularity level $g$ | 3 | | |
| | maximum capacity of thinking memory $N$ | 50 | | |
| | top-$k$ retrieval | 5 | | |

Table 5. Results of multi-image and video based benchmarks on Qwen3-VL-8B.

| Method | MuirBench | | BLINK | | MVBench | | LVBench | | Avarage | |
|---|---|---|---|---|---|---|---|---|---|---|
| | Instruct | Thinking | Instruct | Thinking | Instruct | Thinking | Instruct | Thinking | Instruct | Thinking |
| Single | 64.1 | 75.5 | 68.7 | 64.3 | 68.6 | 68.9 | 55.4 | 58.1 | 64.2 | 66.7 |
| VMAS | 64.5 | 73.0 | 70.3 | 68.4 | 70.2 | 70.8 | 57.5 | 58.4 | 65.6 | 67.6 |
| L$^2$-VMAS | **67.4** ↑2.9 | **77.2** ↑4.2 | **72.7** ↑2.4 | **71.0** ↑2.6 | **72.9** ↑2.7 | **73.1** ↑2.3 | **60.0** ↑2.5 | **61.5** ↑3.1 | **68.2** ↑4.0% | **70.7** ↑4.6% |

agent nodes and task-specific virtual nodes using variational graph autoencoders, generating task-adaptive topologies. To construct the base environments of VMAS, we largely follow the multi-agent settings in (Yu et al., 2025c; Qian et al., 2025).

### D.2. Additional Results

**Analysis of Potential Ineffectiveness.** The slight accuracy drops observed in three cells of Table 1 and Table 2 (*i.e.*, LLaVA-OV-1.5-8B on MMBench benchmark and Qwen3-VL-2B-Thinking on MMBench and RealWorldBenchQA benchmarks) are plausible under a relatively straightforward benchmark or smaller model: when the baseline VMAS is already close to a performance ceiling, additional collaboration information becomes increasingly redundant, yielding small negative fluctuations. However, the reduction of total token usage is still significant.

Importantly, these degradations are isolated and low-magnitude, whereas the aggregated results still show consistent average gains and substantial token reductions across backbones and model scales, indicating that the proposed dual latent memory remains effective and is especially beneficial once tasks/models move away from saturation and require richer cross-turn information integration.

**Results on Augmented Visual Scenarios.** Furthermore, as listed in Table 5, the results provide compelling evidence that our L$^2$-VMAS method maintains substantial performance superiority over baseline approaches in augmented visual scenarios,

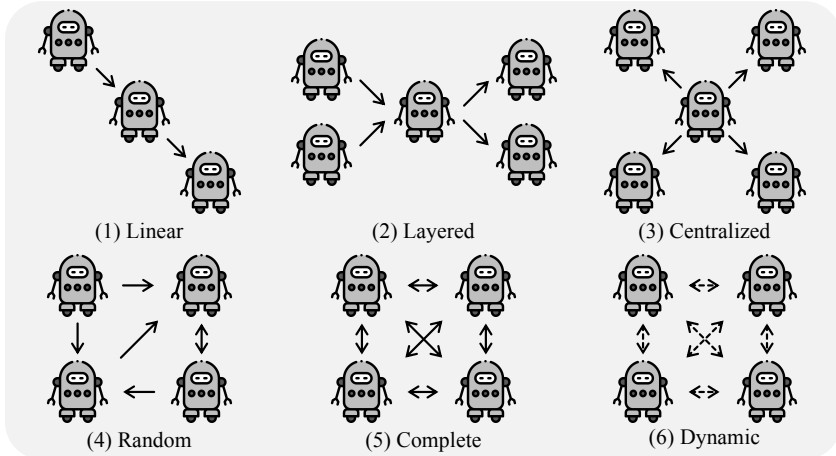

*Figure 9.* The illustrations of the six multi-agent structures in our work.

*Table 6.* Ablation Study on the components of memory synthesis and orchestration.

| Ablation | MMBench | MMStar | RealWorldQA | SimpleVQA |
|---|---|---|---|---|
| *w/o* Triggering | 86.8 (-0.6) | 76.7 (-0.8) | 75.9 (-1.7) | 53.5 (-0.3) |
| *w/o* Attribution | 85.1 (-2.3) | 75.0 (-2.5) | 74.8 (-2.8) | 52.7 (-1.1) |
| *w/o* Perception | 86.6 (-0.8) | 76.2 (-1.3) | 75.1 (-2.5) | 51.9 (-1.9) |
| *w/o* Thinking | 85.3 (-2.1) | 75.2 (-2.3) | 74.4 (-3.2) | 52.3 (-1.5) |
| $L^2$-VMAS | **87.4** | **77.5** | **77.6** | **53.8** |

encompassing both multi-image and video-based tasks. Specifically, when compared with the Single and VMAS baselines, it achieves remarkable improvements across all four benchmark datasets (*i.e.*, MuirBench, BLINK, MVBench, and LVBench) under both Instruct and Thinking settings. On average, the performance gains exceed 4% (4.0% for the Instruct model and 4.6% for the Thinking model), with the most notable advancements observed in MuirBench (Thinking: +4.2%) and LVBench (Thinking: +3.1%), highlighting the effectiveness of our method in addressing diverse visual challenges.

**Effectiveness of Dual Memory System.** As shown in Figure 11, we compare two lines that indicate the triggering rates of perception and thinking memories across perception, thinking, and mixed tasks. The dynamic triggering rates of perception and thinking memories across perception, thinking, and mixed tasks reveal a highly adaptive cognitive mechanism. In perception-focused tasks, perception memory prevails, whereas thinking memory dominates in reasoning-driven scenarios; in mixed tasks, the two engage in a balanced, fluctuating interplay. These patterns demonstrate VMAS's ability to allocate memory resources in alignment with task-specific demands dynamically.

**Scalability.** As demonstrated in Figure 11, we expand the experiments to four benchmarks, which show curves similar to those in Figure 6. Specifically, at the 10*th* turn, our $L^2$-VMAS achieves 6.0-13.9% performance gains based on the Instruct model and 9.2-19.2% on the Thinking model. With improvements surpassing 13% on the RealWorldQA benchmark, it demonstrates enhanced competency and promising potential for tackling sophisticated tasks.

**Ablation & Sensitivity & Generalization Analyses.** As shown in Table 6, the ablation reveals that every component involved in memory synthesis and orchestration plays a non-negligible role in boosting model performance; notably, the removal of Attribution and Thinking induces the most dramatic accuracy declines, whereas the full-component model attains the optimal performance on all benchmarks. As reported in Tables 7, 8, and 9, the sensitivity analyses indicate that our parameter configurations achieve an excellent trade-off between accuracy and efficiency. As shown in Table 10, the proposed $L^2$-VMAS method consistently outperforms the Single and VMAS baselines on unseen datasets, with the most prominent improvement of 6.4% in average accuracy, validating strong cross-task generalization capability.

*Table 7.* Influence of the window length $W$.

| $W$ | MMBench | | MMStar | | RealWorldQA | | SimpleVQA | | Average | |
|---|---|---|---|---|---|---|---|---|---|---|
| | *Accuracy* | *Token* | *Accuracy* | *Token* | *Accuracy* | *Token* | *Accuracy* | *Token* | *Accuracy* | *Token* |
| VMAS | 84.9 | 2190 | 74.8 | 2467 | 74.7 | 2769 | 51.4 | 2492 | 71.4 | 2480 |
| 4 | 85.0 | 1924 | 74.2 | 2117 | 74.2 | 2288 | 51.7 | 2065 | 71.3 | 2099 |
| 8 | 85.3 | 1742 | 75.2 | 1965 | 74.4 | 2133 | 52.3 | 1910 | 71.8 | 1938 |
| 16 | **87.4** | 1682 | **77.5** | 1907 | **77.6** | 2101 | **53.8** | 1865 | **74.1** | 1889 |
| 32 | 85.1 | 1596 | 74.5 | 1914 | 74.5 | 2053 | 52.1 | 1759 | 71.5 | 1831 |

*Table 8.* Influence of the threshold scaling factor $\lambda$.

| $\lambda$ | MMBench | | MMStar | | RealWorldQA | | SimpleVQA | | Average | |
|---|---|---|---|---|---|---|---|---|---|---|
| | *Accuracy* | *Token* | *Accuracy* | *Token* | *Accuracy* | *Token* | *Accuracy* | *Token* | *Accuracy* | *Token* |
| VMAS | 84.9 | 2190 | 74.8 | 2467 | 74.7 | 2769 | 51.4 | 2492 | 71.4 | 2480 |
| 0.3 | 85.3 | 2071 | 75.8 | 2502 | 76.0 | 2716 | 53.3 | 2445 | 72.6 | 2434 |
| 0.4 | **87.6** | 1827 | 77.3 | 2198 | 77.1 | 2438 | **54.0** | 2085 | 74.0 | 2137 |
| 0.5 | 87.4 | 1682 | **77.5** | 1907 | **77.6** | 2101 | 53.8 | 1865 | **74.1** | 1889 |
| 0.6 | 87.2 | 1629 | 76.8 | 1787 | 76.6 | 2065 | 52.9 | 1766 | 73.4 | 1807 |
| 0.7 | 85.2 | 1610 | 75.0 | 1624 | 74.8 | 2029 | 51.3 | 1727 | 71.6 | 1748 |

*Table 9.* Influence of the memory length $L$.

| $L$ | MMBench | | MMStar | | RealWorldQA | | SimpleVQA | | Average | |
|---|---|---|---|---|---|---|---|---|---|---|
| | *Accuracy* | *Token* | *Accuracy* | *Token* | *Accuracy* | *Token* | *Accuracy* | *Token* | *Accuracy* | *Token* |
| VMAS | 84.9 | 2190 | 74.8 | 2467 | 74.7 | 2769 | 51.4 | 2492 | 71.4 | 2480 |
| 2 | 84.4 | 1624 | 74.0 | 1815 | 74.1 | 1990 | 51.8 | 1763 | 71.1 | 1798 |
| 4 | 85.8 | 1656 | 75.0 | 1876 | 74.8 | 2041 | 52.1 | 1817 | 71.9 | 1848 |
| 8 | 87.4 | 1682 | **77.5** | 1907 | **77.6** | 2101 | **53.8** | 1865 | **74.1** | 1889 |
| 16 | **87.6** | 1705 | 76.2 | 1942 | 75.1 | 2169 | 53.7 | 1910 | 73.1 | 1932 |
| 32 | 86.2 | 1790 | 76.1 | 2031 | 75.3 | 2292 | **53.9** | 2013 | 72.9z | 2032 |

*Table 10.* Results of generalization study, which is trained on one dataset and evaluated on unseen benchmarks on Qwen3-VL-8B-Instruct.

| Training | Method | MMBench | | MMStar | | RealWorldQA | | SimpleVQA | | Avarage | |
|---|---|---|---|---|---|---|---|---|---|---|---|
| | | *Instruct* | *Thinking* | *Instruct* | *Thinking* | *Instruct* | *Thinking* | *Instruct* | *Thinking* | *Instruct* | *Thinking* |
| MMBench | Single | - | - | 68.5 | 71.6 | 69.4 | 70.9 | 47.1 | 47.2 | 61.7 | 63.2 |
| | VMAS | - | - | 72.8 | 74.3 | 73.2 | 74.9 | 48.6 | 47.8 | 64.9 | 65.7 |
| | L²-VMAS | - | - | **75.4** ↑2.6 | **76.0** ↑1.7 | **75.9** ↑2.7 | **77.5** ↑2.6 | **49.8** ↑1.2 | **53.4** ↑5.6 | **67.0** ↑3.2% | **69.0** ↑5.0% |
| MMStar | Single | 81.2 | 82.3 | - | - | 68.5 | 70.1 | 47.9 | 47.6 | 65.9 | 66.7 |
| | VMAS | 82.0 | 82.8 | - | - | 70.2 | 73.7 | 48.5 | 49.0 | 66.9 | 68.5 |
| | L²-VMAS | **84.4** ↑2.4 | **85.6** ↑2.8 | - | - | **73.1** ↑2.9 | **76.4** ↑2.7 | **50.4** ↑1.9 | **53.7** ↑4.7 | **69.3** ↑3.6% | **71.9** ↑5.0% |
| RealWorldQA | Single | 80.3 | 82.0 | 66.4 | 70.7 | - | - | 45.8 | 46.0 | 64.2 | 66.2 |
| | VMAS | 81.5 | 82.4 | 68.3 | 71.9 | - | - | 47.1 | 47.1 | 65.6 | 67.1 |
| | L²-VMAS | **84.5** ↑3.0 | **85.7** ↑3.3 | **72.3** ↑4.0 | **75.9** ↑4.0 | - | - | **49.0** ↑1.9 | **52.5** ↑5.4 | **68.6** ↑4.6% | **71.4** ↑6.4% |
| SimpleQA | Single | 81.4 | 82.6 | 68.1 | 71.7 | 69.0 | 70.3 | - | - | 72.8 | 74.9 |
| | VMAS | 82.5 | 83.0 | 72.2 | 73.8 | 73.9 | 74.9 | - | - | 76.2 | 77.2 |
| | L²-VMAS | **84.7** ↑2.2 | **85.9** ↑2.9 | **74.8** ↑2.6 | **75.4** ↑1.6 | **75.6** ↑1.7 | **77.5** ↑2.6 | - | - | **78.4** ↑2.9% | **79.6** ↑3.1% |

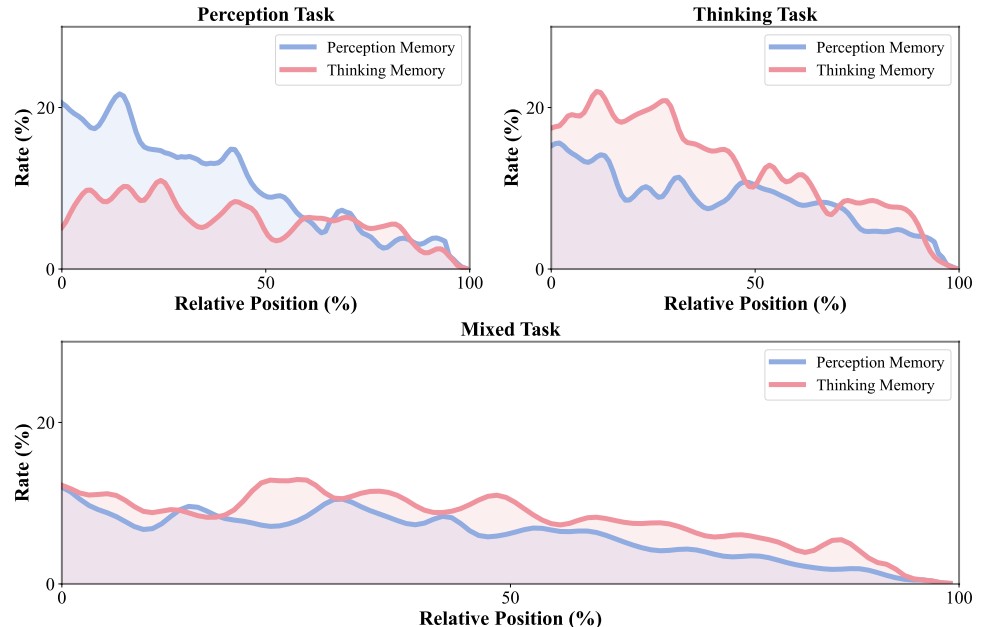

*Figure 10.* Triggering rate of the two memories on perception, thinking, mixed tasks.

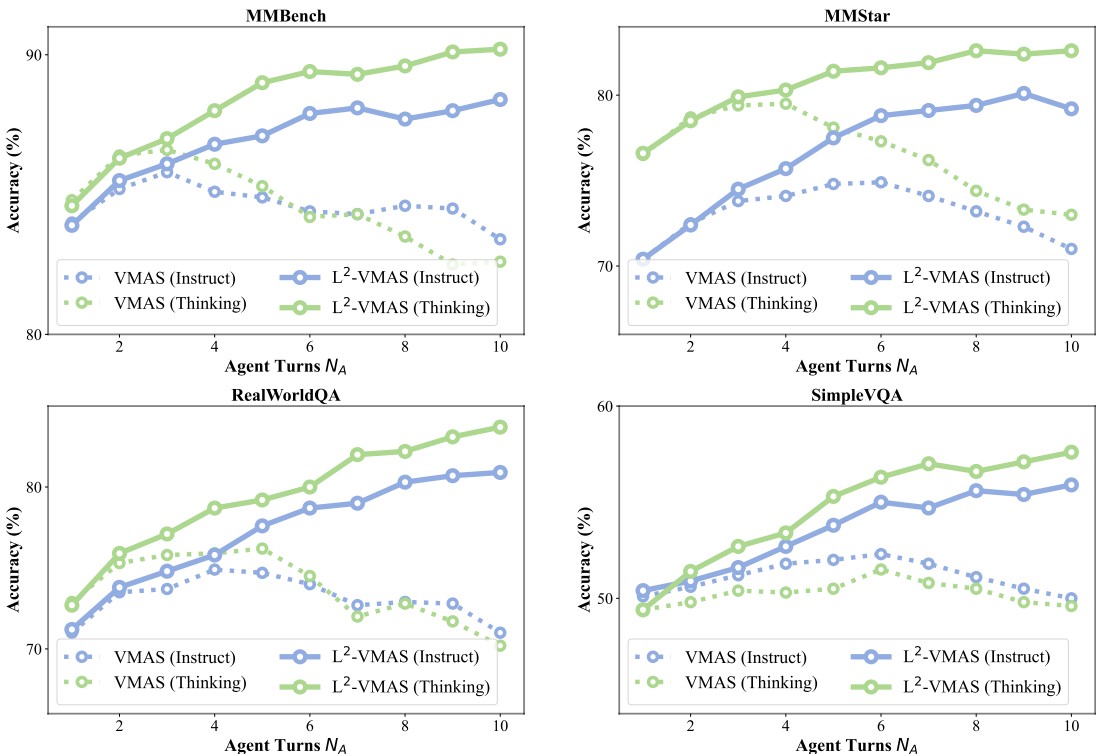

*Figure 11.* Comparison of scalability between VMAS and $L^2$-VMAS based on four benchmarks.

