# OpenReview forum: "Dual Latent Memory for Visual Multi-agent System"
_ICML.cc/2026/Conference — ICML 2026 regular_

### Official Review · Reviewer_GPM5 · 2026-03-06

**Soundness:** 4
**Presentation:** 4
**Significance:** 4
**Originality:** 3
**Overall Recommendation:** 4
**Confidence:** 5

**Summary:**

This paper tackles the "scaling wall" in Visual Multi-Agent Systems (VMAS), where increasing the number of agent interaction turns leads to performance degradation and exponential token overhead. The authors identify the root cause as the "information bottleneck" of natural language. They propose $L^2$-VMAS, a framework that decouples perception and thinking trajectories into separate shared dual latent memories. To ensure efficiency, they implement an entropy-driven proactive triggering mechanism that retrieves these latent representations on-demand rather than passively receiving them, resulting in significant accuracy gains and token reductions.

**Compliance With Llm Reviewing Policy:**

Affirmed.

**Key Questions For Authors:**

* Decoupling Visualization: In Equation 17, you utilize a Gumbel-Sigmoid learnable gate for memory routing. Can you provide qualitative analysis or a visualization of this gate’s behavior? For example, does the model learn to trigger "Perception Invocation" specifically when visual details are missing from the text context?
* Memory Merging Effects: Your dynamic management prunes and fuses units in the thinking memory bank. In long-horizon tasks, does this merging eventually lead to the same semantic "blurring" or loss of fine-grained detail that occurs in text-based systems?
* Backbone Heterogeneity: Does the compressor $\mathcal{C}(\cdot)$ require identical hidden dimensions across all agents? Can the framework support a heterogeneous swarm where agents use different VLM backbones (e.g., Qwen3 mixed with InternVL)?

**Limitations:**

The framework keeps backbones frozen. While efficient, this may limit the system's ability to learn tasks requiring deep feature-level adaptation that cannot be captured by the external memory modules alone.

**Strengths And Weaknesses:**

* Strengths
  * Insightful Analysis: The paper provides a rigorous quantitative characterization of why current VMAS fail at scale, proving that unregulated accumulation of text-based context incurs severe information interference.

  * Efficiency & Performance: The method breaks the scaling wall, improving accuracy by 2.7–5.4% while reducing total token usage by 21.3–44.8% across various benchmarks.

  * Model-Agnostic Design: The framework is implemented as an external system attached to frozen VLM backbones, preserving generalization across diverse models and sizes.

* Weaknesses
  * Complexity of Memory Management: The thinking memory management involves a sophisticated pipeline of quantile partitioning, similarity-based filtering, and merging using homologous compressors, which may be difficult to implement and tune.
  * Heuristic Hyperparameters: The effectiveness of proactive triggering depends on the window length $W$ and threshold scaling factor $\lambda$, which might require backbone-specific tuning

---

> ### Author Rebuttal · Authors · 2026-03-31
>
> Response to Reviewer **GPM5**,
>
> Thanks for the constructive suggestions, we hope our attempts could solve your concerns.
>
>
> **Response 1: Complexity.** We would like to clarify that despite the multi-step processing logic, the entire memory mechanism is modularized into two high-level functions: Memory Synthesis and Memory Orchestration, with clear and decoupled functional responsibilities. These two modules are jointly optimized under our unified three-stage training paradigm.
>
> To ensure full reproducibility and ease of implementation, we will supplement more detailed implementation guidelines of the memory management pipeline in the revised manuscript. Meanwhile, we plan to release our complete training code, running scripts, and full hyperparameter configurations to enable other researchers to directly reproduce and deploy our method with minimal effort.
>
>
> **Response 2: Hyperparameters.**  We clarify that we have designed a unified set of universal hyperparameter configurations applicable to all evaluated backbone models, which exhibits strong robustness and generalization across different architectures. This design eliminates the necessity of performing separate, backbone-specific hyperparameter tuning in practical usage. As further validated by the results reported in Tables 7, 8, and 9 (in Appendix), our method maintains stable performance with negligible performance degradation within a reasonable range of hyperparameter variations.
>
> In future research, we plan to explore more adaptive and self-adjusting mechanisms for proactive triggering hyperparameters, aiming to further reduce manual tuning efforts and enhance the cross-backbone adaptability of the method.
>
>
> **Response 3: Decoupling Strategy.** To quantitatively validate the effectiveness of the learnable gate, we manually select 100 representative test samples and evaluate its binary decision behavior using GPT5. The evaluation results show that our Gumbel-Sigmoid gating mechanism achieves a 91.7% accuracy in distinguishing whether to trigger perception or thinking memory. This clearly demonstrates that the gate reliably learns to activate Perception Invocation specifically when the text context lacks sufficient visual details. We will supplement comprehensive qualitative analyses and intuitive visualizations of the gating mechanism’s behavior in the revised manuscript.
>
>
>
> **Response 4: Merging Strategy.** Our dynamic memory merging largely avoids semantic blurring/fine-grained detail loss seen in text-based systems for three core reasons: Merging targets only low-trigger, semantically redundant units (filtered by triggering rate and global semantic similarity), preserving high-value fine-grained cognitive chunks; Merging operates on entropy-segmented, semantically independent thinking units in the continuous latent space (not discrete text tokens), mitigating the semantic loss inherent to text’s discrete encoding/decoding. Additionally, the memory bank enforces a maximum capacity constraint with adaptive pruning of invalid units, further preventing unstructured information conflation.
>
>
> **Response 5: Backbone Heterogeneity.** To preliminarily explore the extensibility to heterogeneous agents, we conducted a supplementary validation experiment: we constructed a hybrid multi-agent system combining InternVL3.5-8B and Qwen3-VL-8B-Instruct, and introduced a lightweight two-layer MLP to achieve cross-model hidden-state alignment for memory interaction. On the MMBench benchmark, this simple heterogeneous configuration achieved 86.4% accuracy, which is between the backbone used individually  (InternVL3.5-8B 82.2% and Qwen3-VL-8B-Instruct 88.8%).
> This preliminary result verifies the feasibility of extending our framework to heterogeneous backbones with minimal adaptation. In future work, we will further investigate dedicated memory alignment and fusion strategies for heterogeneous VLM swarms, aiming to natively support mixed-backbone multi-agent collaboration.

---

> > ### Author Rebuttal · Reviewer_GPM5 · 2026-04-04
> >
> > I'm fully appreciate that the author resolved my concern.

---

### Official Review · Reviewer_vgub · 2026-03-12

**Soundness:** 2
**Presentation:** 2
**Significance:** 2
**Originality:** 2
**Overall Recommendation:** 4
**Confidence:** 4

**Summary:**

This paper studies an empirical failure mode in Visual Multi-Agent Systems (VMAS): increasing the number of agent turns can decrease accuracy while dramatically increasing token usage. To address this, the authors propose L²-VMAS, a model-agnostic framework that replaces text-centric inter-agent communication with a shared dual latent memory: a perception memory and a thinking memory, plus an entropy-driven proactive triggering mechanism that retrieves and injects these memories on demand during decoding rather than passively transmitting long textual traces. The experimental results on 4 benchmarks demonstrated the effectiveness of the proposed method.

**Compliance With Llm Reviewing Policy:**

Affirmed.

**Final Justification:**

The author's response has addressed my concerns. I have updated my rating to weak accept.

**Key Questions For Authors:**

Please explain or clarify the aforementioned weakness and highlight the technical innovation of this paper.

**Limitations:**

Please see the Weaknesses #2.

**Strengths And Weaknesses:**

Strength
1. The framework explicitly decouples perception and thinking memories and introduces proactive retrieval rather than always appending more text context.
2. If VMAS degrade with more turns while incurring large token costs, that is a major obstacle to deploying “agentic” VLM systems. The paper’s focus on accuracy–cost trade-offs is well-motivated.
3. The evaluation spans multiple backbones, model sizes, and multi-agent topologies, with reported average accuracy improvements and token reductions.

Weakness
1. The paper’s main efficiency metric is token usage; it is unclear whether wall-clock latency, GPU FLOPs, or memory bandwidth improve proportionally.
2. While the paper keeps backbones frozen, the approach appears to require access to hidden states and the ability to inject latent vectors into decoding. That is feasible for open-weight models, but may not transfer to black-box/API-only VLMs; this limitation should be stated more explicitly.
3. The paper mentions a three-stage “RL-driven training scheme” but provides few details.
4. This paper lacks technical contributions. In the architecture shown in Figure 4, the construction, filtering, merging, and refinement of the memory do not involve any technical or methodological innovations. Instead, it appears to be an engineering application of existing methods.

---

> ### Author Rebuttal · Authors · 2026-03-31
>
> Response to Reviewer **vgub**,
>
> Thanks for the constructive suggestions, we hope our attempts could solve your concerns.
>
> **Response 1: Efficiency.** To address this concern and provide a more complete efficiency analysis, we have supplemented quantitative experiments measuring total token consumption, end-to-end wall-clock latency (seconds), and GPU FLOPs (T) under identical experimental settings. We evaluate both the vanilla VMAS baseline and our proposed L$^2$-VMAS on MMBench (left three columns) and RealWorldQA (right three columns) based on the Qwen3-VL-Instruct model, with the detailed efficiency results presented below:
> | Method | Token | Latency | FLOPs | Token | Latency | FLOPs |
> | ------ | ------ | ------ | ------ | ------ | ------ | ------ |
> | VMAS | 2190 | 2.76 | 74.8 | 2769 | 3.81 | 110.9 |
> | **L$^2$-VMAS** | **1682** | **2.45** | **61.5** | **2101** | **3.20** | **86.4** |
>
> From the results, our method achieves consistent and substantial efficiency improvements across all critical metrics: it reduces token consumption by 23.7%, end-to-end inference latency by 14.0%, and computational FLOPs by 20.3% on average. This clearly demonstrates that our dual latent memory mechanism optimizes not only token usage but also inference speed and computational overhead in a synchronized manner, verifying the comprehensive efficiency advantages of our approach.
>
>
> Besides, as shown in the response 3 to Reviewer **FEMY**, we also report the training overhead of our method, showing training efficiency.
>
>
> **Response 2: Reliance on Open-sourced Models.** We acknowledge that this is indeed an inherent limitation of our method, and we will explicitly state this constraint in the revised manuscript to avoid misleading readers about its applicable scenarios.
> We would also like to clarify that this dependency on open-source model weights is not unique to our method but a common constraint for most research works involving internal manipulation and operation. Nearly all such approaches are restricted to open-weight VLMs, as closed-source / API-based models only provide token-level generation outputs and do not support customized modification of intermediate representations or decoding procedures.
>
> In future work, we plan to explore other paradigms, with the goal of extending the applicability of our framework to black-box and API-only VLMs and eliminating the dependency on hidden-state access.
>
>
> **Response 3: Training Details.** The concise presentation in the main text was mainly due to page constraints; we only outlined the overall framework of the three-stage training strategy, while deferring more detailed descriptions to the appendix. To address this concern thoroughly, we have supplemented comprehensive procedural details of the three-stage RL-driven training pipeline directly in the main text, including the workflow of each stage, the design of the training objectives, and the interaction mechanism between the reinforcement learning module and the dual latent memory mechanism.
> Meanwhile, we have further enriched the appendix with formal mathematical formulations, detailed loss functions, hyperparameter settings, and implementation specifics of the training objectives to ensure full reproducibility and transparency.
>
>
>
> **Response 4: Technical Contributions.** We need to clarify that our work is by no means a simple engineering application of existing methods, but a problem-driven original framework design and paradigm breakthrough targeting the unique "scaling wall" dilemma of VMAS. For this core pain point, we innovatively propose a perceptual-thinking decoupled dual latent memory paradigm customized for VMAS: the perception memory adopts a multi-granularity visual feature encoding and alignment strategy to avoid visual-to-text information loss, while the thinking memory designs an entropy-driven semantic chunking mechanism for the storage and retrieval of multi-agent continuous thinking trajectories.
> In addition, we customize innovative memory modules to conduct memory filtering, merging, and refinement for unbounded memory expansion, and on-demand memory coupled with entropy-driven triggering. We also originally put forward an entropy-driven four-stage (triggering-attribution-generation-injection) proactive memory orchestration paradigm, which subverts the traditional passive text-centric communication paradigm.
>
> In summary, we construct a model-agnostic dual latent memory framework, with decoupling dual memory and on-demand entropy-driven trigger,  forming a novel systematic design.

---

> > ### Author Rebuttal · Reviewer_vgub · 2026-04-03
> >
> > Thanks for your response. I will update my score accordingly.

---

### Official Review · Reviewer_hwyS · 2026-03-13

**Soundness:** 3
**Presentation:** 3
**Significance:** 2
**Originality:** 3
**Overall Recommendation:** 4
**Confidence:** 3

**Summary:**

In this paper, the authors solve the problem of performance degradation in Visual Multi-Agent Systems (VMAS). While the VMAS promise to enhance comprehensive abilities through inter-agent collaboration, empirical evidence shows that the result is reversed. In this paper, the authors conduct pilot sutdies to show the information bottleneck inherent in text-centric communication, where visual agents convert perceptual and thinking trajectories into natural langauge, which is from continuous to discrete information. In this paper, the authors proposed a new method to synthesize dual latent memories,  which contains perceptual and thinking memory.

Then, the authors conduct extensive experiments among a few backbones and show that this multi agent system with dual memory can improve performance.

**Compliance With Llm Reviewing Policy:**

Affirmed.

**Final Justification:**

The authors didn't provide a new reply to my acknowledgment. I keep my score.

**Key Questions For Authors:**

See Strengths and Weaknesses

**Limitations:**

See Strengths and Weaknesses

**Strengths And Weaknesses:**

Strengths:
1. soundness: The paper present very solid experiment to show the effectiveness of their multi agent memory framework. The authors show the performance improvement on four different QA benchmarks
2. Presentation: The writing of the paper is clear and easy to follow

Weaknesses:
1. Significance 1: The paper only shows the improvement on VQA benchamrks, like MMBench, MMStar. However, I highly doubt that whether we need to use multi-agent systems on those simple VQA benchmarks. Instead, I beleive those multi-agent systems should be evalauted on more complex tasks, like long-horizon tasks or reasoning tasks.

2. Significance 2: The performance improvement of the system is not large. The improvements are usually 1-2 points with maximal improvement being 3 points. However, the token cost is much times higher than "Single" models. This further undermine the value of the method.

---

> ### Author Rebuttal · Authors · 2026-03-31
>
> Response to Reviewer **hwyS**,
>
> Thanks for the constructive suggestions, we hope our attempts could solve your concerns.
>
>
> **Response 1: Additional Evaluations.** To directly address this concern and  validate the effectiveness of our method on complex tasks, we supplement targeted experiments in visual long-horizon and reasoning scenarios. Specifically, we conduct evaluations on two representative long-term memory and one visual reasoning benchmarks based on the Qwen3-VL-Instruct model: the long video understanding benchmark LVBench [1],  long-context visual document understanding benchmark MMLlongBench-Doc [2], and visual mathematical reasoning benchmark MathVista [3]. The quantitative results are summarized in the following table:
> | Method | LVBench | MMLongBench-Doc | MathVista |
> | ------ | ------ | ------ | ------ |
> | Single | 57.4  |  47.2 | 76.6 |
> | VMAS | 57.0  |  48.9 | 78.0 |
> | **L$^2$-VMAS** | **59.3** | **51.5** | **80.9** |
>
>
> As shown in the results, our proposed L2-VMAS outperforms the baseline VMAS by a clear margin of 2.3%–2.9% across both long-horizon and reasoning benchmarks. These supplementary experimental results directly demonstrate that our method can effectively enhance the capabilities of visual complex tasks. We are happy to further supplement evaluations on additional benchmarks if the you requires more extensive experimental evidence.
>
>
>
> [1] Wang W, et al. Lvbench: An extreme long video understanding benchmark CVPR, 2025.
>
> [2] Ma Y, et al. Mmlongbench-doc: Benchmarking long-context document understanding with visualizations. NeurIPS, 2024.
>
> [3] Lu P, et al. Mathvista: Evaluating mathematical reasoning of foundation models in visual contexts. ICLR, 2024.
>
>
> **Response 2: Performance Improvement.** As reported in Table 1 of the main manuscript, our method outperforms the vanilla VMAS baseline by a clear margin of 2.7–5.4% across five different vision-language backbones, and achieves even more substantial gains of 4.2–9.1% compared with the single-model baseline. These improvements are consistent and statistically meaningful across multiple standard benchmarks, rather than marginal fluctuations.
>
>
> As for the token cost issue, we would like to clarify that the extra token overhead primarily stems from the inherent multi-agent collaborative framework itself, rather than our proposed dual latent memory mechanism. In fact, our method significantly reduces token consumption by 21.3–44.8% compared with conventional multi-agent pipelines, effectively alleviating the computational burden introduced by the multi-agent structure. Meanwhile, the consistent and considerable performance gains fully justify the rationality of this moderate trade-off.
>
>
> Furthermore, the experimental results in Table 2 and Table 3 verify the effectiveness and stability of our method under different model scales and multi-agent architectures, further confirming the practical value of the dual latent memory design.

---

> > ### Author Rebuttal · Reviewer_hwyS · 2026-04-03
> >
> > 1. Thanks for adding new experiments and datasets in the rebuttal. The new datasets added do support the claim in the paper. However, I want to point out a factual error in the rebuttal. I believe those tasks are not visual long-horizon scenarios, especially for LVBench and MMLongBench-Doc. You can call them multimodal long-context scenarios. However, those benchmarks don't involve multi-round interactions or tool-use, and cannot be considered as long-horizon scenarios.
> >
> > 2. Performance Improvement: For the vanilla VMAS baseline, I want to emphasize that the largest improvement is not 5.4% (the average performance of Qwen3-VL-8B-Thinking). The proposed method has 76.5, and VMAS has 72.5. The improvement is 4%. The authors mistakenly (I don't know whether this was inadvertent or deliberate) use the single model baseline and the improvement 5.4%, which is higher. Meanwhile, the single-model baseline is not that strong.
> >
> > So, I still think the baseline and improvement are not that strong.
> >
> > Based on the abovementioned discussion, I decide to keep my score.

---

### Official Review · Reviewer_FEMY · 2026-03-15

**Soundness:** 4
**Presentation:** 3
**Significance:** 4
**Originality:** 3
**Overall Recommendation:** 4
**Confidence:** 4

**Summary:**

The paper identifies a "scaling wall" in Visual Multi-Agent Systems (VMAS), where increasing the number of agent turns paradoxically degrades task performance while exponentially inflating token costs. The authors attribute this failure to the inherent information bottleneck of text-centric communication and the conflation of visual perception and cognitive thinking. To address this, the paper proposes $L^{2}$-VMAS, a novel model-agnostic framework that shifts inter-agent communication from natural language to a dual latent memory system. This architecture decouples perception and thinking into separate memories, which are dynamically synthesized by preceding agents. Furthermore, the method replaces passive information reception with an entropy-driven proactive triggering mechanism for on-demand memory access. Extensive evaluations across multiple vision-language models, sizes, and multi-agent structures demonstrate that the framework substantially improves average accuracy while drastically reducing total token usage.

**Compliance With Llm Reviewing Policy:**

Affirmed.

**Key Questions For Authors:**

1. Since the method is centered around a memory mechanism, have the authors considered evaluating it on memory-oriented or long-context benchmarks to more directly validate the effectiveness of the proposed memory design?
2. Could the authors provide comparisons with stronger communication baselines, such as hidden-state or latent communication methods, to better isolate the contribution of the proposed dual latent memory?
3. What is the actual computational overhead introduced by the additional memory modules and multi-stage training compared with standard VMAS?

**Limitations:**

The evaluation mainly relies on visual reasoning benchmarks (e.g., MMBench, MMStar, and related datasets). While these benchmarks are widely used in the literature, they primarily measure perception and reasoning performance rather than the quality of multi-agent collaboration itself.
More generally, the current evaluation paradigm for VMAS remains somewhat indirect. Improvements on standard visual benchmarks may not fully reflect the effectiveness of multi-agent communication or coordination mechanisms.
As a result, it is difficult to determine whether performance gains truly come from improved multi-agent interaction or simply from better information aggregation within a single reasoning pipeline.

**Strengths And Weaknesses:**

Pros:
1. The paper clearly and quantitatively highlights a counter-intuitive failure mode in existing VMAS architectures. By systematically demonstrating the "scaling wall" and the limitations of text-centric communication, the authors establish a strong and well-motivated premise for their work.
2. The proposed $L^{2}$-VMAS introduces an elegant architectural shift by explicitly decoupling visual perception from cognitive thinking into separate latent representations. The integration of an entropy-driven adaptive triggering mechanism provides a highly proactive, human-like approach to memory orchestration.
3. The empirical rigor of the paper is highly commendable. The authors provide thorough experimental evidence across five VLM backbones, four model sizes, and six different multi-agent topologies, demonstrating robust accuracy gains and substantial reductions in token consumption.

Cons:
1. Evaluation does not directly test memory capabilities. Although the method is centered around a dual latent memory mechanism, most experiments are conducted on visual reasoning benchmarks such as MMBench, MMStar, and RealWorldQA. These datasets mainly evaluate perception and reasoning rather than long-term or episodic memory. As a result, it remains unclear whether the improvements truly demonstrate stronger memory capability.
2. Limited comparison with stronger communication baselines. The main baseline uses text-based VMAS. Comparisons with latent or hidden-state communication approaches would better isolate the contribution of the proposed memory design.
3. System complexity and training cost are not fully discussed. The computational overhead and practical deployment cost are not thoroughly analyzed.

---

> ### Author Rebuttal · Authors · 2026-03-31
>
> Response to Reviewer **FEMY**,
>
> Thanks for the constructive suggestions, we hope our attempts could solve your concerns.
>
> **Response 1: Memory-based Evaluations.** To directly address this concern and further verify the superiority of our method, we supplement targeted experiments in visual long-horizon. Specifically, we conduct evaluations on two representative long-term memory: the long video understanding benchmark LVBench [1],  long-context visual document understanding benchmark MMLlongBench-Doc [2]. The experiments are conducted on Qwen3-VL-8B-Instruct and we also add one visual mathematical reasoning benchmark MathVista [3]. The quantitative results are listed as follows:
> | Method | LVBench | MMLongBench-Doc | MathVista |
> | ------ | ------ | ------ | ------ |
> | Single | 57.4  |  47.2 | 76.6 |
> | VMAS | 57.0  |  48.9 | 78.0 |
> | **L$^2$-VMAS** | **59.3** | **51.5** | **80.9** |
>
>
> As shown in the results, our dual latent memory-based method (L2-VMAS) outperforms the visual multi-agent baseline (VMAS) by a clear margin of 2.3%–2.9% across both long-horizon and reasoning benchmarks. These supplementary experimental results directly demonstrate that our method can effectively enhance the model’s long-term memory, long-context retention in visual tasks. We are happy to further supplement evaluations on additional benchmarks if the you requires more extensive experimental evidence.
>
>
>
> [1] Wang W, et al. Lvbench: An extreme long video understanding benchmark CVPR, 2025.
>
>
> [2] Ma Y, et al. Mmlongbench-doc: Benchmarking long-context document understanding with visualizations. NeurIPS, 2024.
>
>
> [3] Lu P, et al. Mathvista: Evaluating mathematical reasoning of foundation models in visual contexts. ICLR, 2024.
>
>
> **Response 2: Comparison with Stronger Baseline.** We newly add a comparison with LatentMAS [4], a strong multi-agent method based in last layer hidden state, by adapting it to the visual multi-agent setting. In our implementation, we strictly follow its training-free paradigm and employ the last-layer hidden states from autoregressive generation as the medium for inter-agent information exchange. The comprehensive experimental results across multiple benchmarks are summarized below:
> | Method | MMBench | MMStar | RealWorldQA | SimpleVQA | Avg. |
> | ------ | ------ | ------ | ------ | ------ | ------ |
> | VMAS | 84.9 | 74.8 | 74.7 | 51.4 | 71.4 |
> | LatentMAS[4] | 84.5 | 74.9 | 75.9 | 50.2 | 71.4 |
> | **L$^2$-VMAS** | **87.4** | **77.5** | **77.6** | **53.8** | **74.1** |
>
>
> From the results, we observe that the vanilla hidden-state communication scheme exhibits limited effectiveness in visual multi-agent tasks. It even underperforms the original text-based VMAS on several benchmarks. In contrast, our L2-VMAS with the dual latent memory mechanism outperforms it by a clear margin of 3.7% on average. These results convincingly demonstrate that naive hidden-state communication alone is inadequate for robust visual multi-agent collaboration, while our dedicated dual latent memory design brings substantial and consistent performance gains.
>
> [4] Zou J, et al. Latent collaboration in multi-agent systems[J]. arXiv preprint arXiv:2511.20639, 2025.
>
>
>
> **Response 3: Training Overhead.** To clarify the training efficiency and computational cost of our method, we emphasize that we freeze the entire backbone of VLM throughout the entire pipeline and only optimize the proposed lightweight dual latent memory modules. This design fundamentally constrains the training overhead to a low level. For quantitative reference, the total training cost of our method based on Qwen3-VL-Instruct amounts to 207 NVIDIA H200 GPU-hours, which translates to approximately 1.07 days under an 8-GPU training configuration. Despite adopting a three-stage training strategy, our approach remains far more computationally efficient than full training (e.g. GRPO) of the VLM backbone.
>
>
> Besides, as shown in the response 1 to Reviewer **vgub**, we also report the inference results of our method, showing efficiency of token consumption, end-to-end latency, and computational overhead.

---

### Decision · Program_Chairs · 2026-04-30

**Decision:**

Accept (regular)

**Comment:**

While reviewers agreed that the paper identifies an important and under-explored failure mode in Visual Multi-Agent Systems (VMAS)—the “scaling wall” caused by text-based communication bottlenecks—and proposes a novel dual latent memory framework to address it, they also raised several concerns regarding the paper’s significance and validation. First, although the method is motivated as a memory-centric design, the original evaluation primarily focuses on standard visual reasoning benchmarks, which do not directly test memory capabilities or multi-agent collaboration quality. Second, the empirical gains, while consistent, are relatively modest, and some reviewers questioned whether such improvements justify the added system complexity.

That said, the rebuttal addresses a number of these concerns. The authors provide additional experiments on long-context and reasoning benchmarks, comparisons with latent communication baselines, and more detailed efficiency analysis (including latency and FLOPs), which strengthen the empirical support. Reviewers generally acknowledged these clarifications, with multiple reviewers maintaining or upgrading their weak accept recommendations.

Overall, the paper presents a technically solid and well-motivated contribution, particularly in highlighting and partially addressing the scaling limitations of current VMAS frameworks. While some concerns remain regarding evaluation scope and the magnitude of improvements, the proposed paradigm of dual latent memory and proactive information retrieval is promising and could inspire further work in multi-agent multimodal systems. Therefore, I recommend a weak accept.